# Structural domain in the Titin N2B-us region binds to FHL2 in a force-activation dependent manner

Yuze Sun [1], Xuyao Liu[2], Wenmao Huang [2], Shimin Le[2] & Jie Yan [1,2,3,4] ✉

Titin N2B unique sequence (N2B-us) is a 572 amino acid sequence that acts as an elastic spring to regulate muscle passive elasticity. It is thought to lack stable tertiary structures and is a force-bearing region that is regulated by mechanical stretching. In this study, the conformation of N2B-us and its interaction with four-and-a-half LIM domain protein 2 (FHL2) are investigated using AlphaFold2 predictions and single-molecule experimental validation. Surprisingly, a stable alpha/beta structural domain is predicted and confirmed in N2B-us that can be mechanically unfolded at forces of a few piconewtons. Additionally, more than twenty FHL2 LIM domain binding sites are predicted to spread throughout N2B-us. Single-molecule manipulation experiments reveals the force-dependent binding of FHL2 to the N2B-us structural domain. These findings provide insights into the mechano-sensing functions of N2B-us and its interactions with FHL2.

Titin is the largest protein in the human body, with a size ranging from 3.4 to 3.9 MDa[1]. It spans from the Z disk to the M band region of the sarcomere and has essential structural and signalling functions in contractile units. The gene encoding titin, TTN, is expressed in three main isoforms in human adults, with the N2B and N2BA isoforms being expressed in the heart, and the N2A isoform in skeletal muscles. The selective expression of these different isoforms has a significant impact on cardiac functions[2,3].

In the I-band of the sarcomere, titin has an elastic spring-like component that connects the thick filament and Z-disk. This component acts as a molecular spring, playing a crucial role in the assembly and maintenance of the sarcomere ultrastructure. It is composed of tandem immunoglobulin (Ig) domain repeats and three intrinsically disordered regions, namely the PEVK region, N2A-us region, and N2B-us region. These three regions play differential roles in regulating titin's functions in the sarcomere[4–6].

Despite the mechanism remaining unknown, the N2B element plays an important role in the pathogenesis of cardiac diseases. Targeted deletion of N2B element in mouse leads to diastolic dysfunction and cardiac atrophy[7], and N2B-KO mouse has been used widely as a research animal model for cardiac diseases[8–10].

N2B-us is often thought to be intrinsically disordered and acts as an elastic spring[11]. Its flexibility can be modulated through phosphorylation by ERK2, CaMKIIδ[12], PKG[13], and PKA[14]. As a flexible force-bearing element in titin, the conformation of N2B-us is likely regulated by forces in the piconewton range[15].

Furthermore, N2B-us has been found to interact with a number of signalling proteins, including Erk2[16], FHL1[17], FHL2[18], and $\alpha/\beta$-crystallin[19], and serves as a signalling hub with a significant impact on cardiomyocyte function. Dysregulated interactions between these factors and N2B-us have been linked to diseases such as cardiac hypertrophy[20]. Given its role as a force-bearing component within titin, one could hypothesize that the interactions between N2B-us and these signalling proteins might be influenced by the mechanical stretching of N2B-us during cardiac cycles. The force-dependence of the interactions between N2B-us and most of its binding partners remains largely unexplored.

Among these N2B-us interacting signalling proteins, FHL2 (four and a half LIM protein-2) is a critical protein that is abundantly

[1]Mechanobiology Institute, National University of Singapore, Singapore, Singapore. [2]Department of Physics, National University of Singapore, Singapore, Singapore. [3]Centre for Biological Imaging Sciences, National University of Singapore, Singapore, Singapore. [4]Joint School of National University of Singapore and Tianjin University, International Campus of Tianjin University, Binhai New City, Fuzhou, China. ✉e-mail: phyyj@nus.edu.sg

expressed in the heart and has a profound effect on heart function. Mutations in FHL2 are associated with human cardiac hypertrophy[21]. FHL2-deficient mice show no abnormalities in heart development but have a serious lack of ability to recover from various types of cardiac insults[22,23]. FHL2 binds to N2B-us, acting as a scaffold to recruit metabolic enzymes to titin[18] and negatively regulates ERK2[24]. In addition, FHL2 plays an important role in regulating the expression of several genes in cardiac tissue[25]. However, little is known about the FHL2 binding sites in N2B-us, the conformation of the bound complex, and most importantly, how the binding is regulated by physiologically relevant forces.

To investigate these questions, we combined the use of AlphaFold2 structural prediction and single-molecule manipulation experiments. This allowed us to identify a thermodynamically stable 115 a.a. structured domain in N2B-us with a melting temperature of ~62 °C and a high folding energy of ~−8 kcal/mol. This domain unfolds at physiological levels of force, a few piconewtons (pN), at around 37 °C. Using the same approach, we were able to predict 24 LIM domain binding sites of FHL2 throughout N2B-us, including eight cryptic sites within the identified structural domain. This prediction was confirmed by single-molecule experiments for FHL2's binding to the disordered region and the region bearing the structural domain. Importantly, the binding of FHL2 to the predicted cryptic sites in the structural domain requires the mechanical unfolding of the domain. Furthermore, the binding of FHL2 to N2B-us leads to multiple force-dependent bound conformations.

## Results

### N2B-us contains a stable structural domain

To probe the conformation of N2B-us, we input the 572 a.a. N2B-us sequence to AlphaFold2[26]. Surprisingly, a structural domain of about 115 a.a. was predicted with high confidence, spanned between disordered regions that lack tertiary structures. The unexpected structural domain predicted by AlphaFold2 contains three beta strands spanned between two alpha-helical regions (Figs. 1A and S1).

To verify the existence and localization of such a structural domain in N2B-us, we prepared a protein construct where the sequence of the N2B-us region is inserted between four well-characterized titin Ig 27th domains (I27, two repeats at each end) acting as a molecular spacer[27]. The N- and C- termini of the construct contain a biotinylated AviTag[28] and a SpyTag[29], respectively, for specific tethering to a coverslip surface and a superparamagnetic microbead (M270, Invitrogen; Fig. 1B, right panel). Time-varying or constant external forces were applied to tethered protein constructs via the end-attached microbead using an in-house built magnetic-tweezers setup[30,31]. The height of the microbead was recorded in real-time at a nanometre resolution. Due to force balance, the tension in the molecule is the same as the applied force. In addition, a stepwise bead height change at a constant or a time-varying force is the same as the extension change of the molecule[31]. The force was calibrated based on extrapolation with a standard curve, which is associated with 10% relative error as described in our previous publications[30,31]. The N2B-us region contains six cysteine residues. 10 mM DTT was included in the buffer solution to prevent formation of disulfide bonds[32].

Figure 1B left panel shows a typical force-height curve of full-length N2B-us construct during a force-increase scan with a force-loading rate of 1 pN/s from 1 pN at 23°C. A stepwise bead height increase was observed at a force around 10 pN, indicating a stable compact element in N2B-us. To see whether this compact element corresponds to the AlphaFold2-predicted structural domain in N2B-us, a construct with only the sequence of 115a.a. structural domain (N2B-us-S) predicted by AlphaFold2 was expressed and purified. A similar stepwise bead height increase was observed when the magnetic tweezer experiment was repeated on N2B-us-S (Fig. 1C). However, when the experiment was repeated on a construct containing the N2B-

us sequence with the predicted structural domain deleted (N2B-us-ΔS), the stepwise bead height increase disappeared (Fig. 1D). More representative traces for each construct were provided in Figs. S2–S4.

These results confirm the existence of the predicted structural domain in N2B-us, which could withstand a few pN forces. Several papers have shown that N2B us contains six cysteines that can form disulfide bonds in an oxidative environment[32–34]. The observed highly characteristic unfolding signal is unlikely from the formation of disulfide bonds due to the presence of DTT. Further, should the disulfide bonds form, they cannot be unfolded by pN forces as covalent bonds typically require nanonewton forces to break[35]. Additionally, based on the structural prediction with AlphaFold2, the three cysteines in the N2B-us structural domain are distal from each other, suggesting that they are unlikely to form disulfide bond once the domain is folded into the structure even under conditions that allow formation of disulfide bonds (Fig. S5). This was experimentally confirmed by repeating the experiments in PBS buffer without DTT where the same unfolding signal was observed (Fig. S6).

The structural domain undergoes reversible unfolding and refolding fluctuations with nearly equal probabilities around a critical force of $F_c = 5.5 \pm 0.5$ pN at room temperature(23°C) (Fig. 2A). The time fractions of the unfolded and the folded states are proportional to the probabilities of the corresponding states. The average ratio of the time fractions from data obtained from four independent tethers (Fig. S7) with total time more than 3000 s was determined to be $\frac{p_{fold}}{p_{unfold}} \approx 1.0$, from which the zero-force folding energy was calculated to be $\Delta G_0 = -13.0 \pm 1.9 K_B T$ using the equation:

$$\Delta G_0 = -k_B T \ln\left(\frac{p_{fold}}{p_{unfold}}\right) + \int_0^F (x_0(f) - x_u(f))df \qquad (1)$$

where $x_0(f)$ and $x_u(f)$ are the force-extension curves of the folded and unfolded states of the domain (Supplementary note 1). The domain retains a similar mechanical stability at 37°C, indicating that it could be sensitively regulated by mechanical force at physiological relevant temperature (Fig. S8). The error bar in the estimated $\Delta G_0$ was obtained by bootstrap analysis and error propagation calculation (Supplementary note 2). The $\Delta G_0$ was estimated based on assuming the unfolded state as a randomly coiled polymer with a persistence length of 0.8 nm, which might cause certain uncertainty as the persistence length of a peptide polymer may vary over a range from 0.4 nm to 0.8 nm[36–38]. The estimated value of $\Delta G_0$ is comparable or greater than several known stable protein domains such as titin I27 Ig domain and α− actinin spectrin repeats[39].

The data from the single-molecule manipulation experiments strongly suggest that the domain has a stable folded structure. To further confirm its existence and thermal stability, we conducted a protein thermal shift assay using the GloMelt™ Thermal Shift Protein Stability Kit from Biotium. A distinct single peak was observed at ~62 °C (Fig. 2B), indicating that a structurally stable domain melted at this temperature, which is similar to the melting temperature of I27 (~70 °C)[40,41] and a goat IgG domain measured in this study as a control (Fig. S9). Together, these results confirm that the predicted structural domain by AlphaFold2 is a thermally stable structure, yet a mechanosensitive domain that can be unfolded by pN forces, in N2B-us.

Next, we want to investigate if the unfolding step is consistent with the predicted structure. We recorded the unfolding forces and step sizes in more than 100 times of unfolding events for both N2B-us-S and full-length N2B-us constructs (Fig. 2C–J). The difference between $x_0(f)$ and $x_u(f)$ is the theoretical force-dependent step size (dashed curve in Fig. 2D, H, Supplementary note 1). The predicted curve is consistent with experimental data, suggesting that the steps are due to the unfolding of the predicted structural domain. The unfolding force distribution of N2B-us-S and full-length N2B-us was quantified and fitted with a Gaussian distribution; the mean values are 10.7 ± 1.1 pN

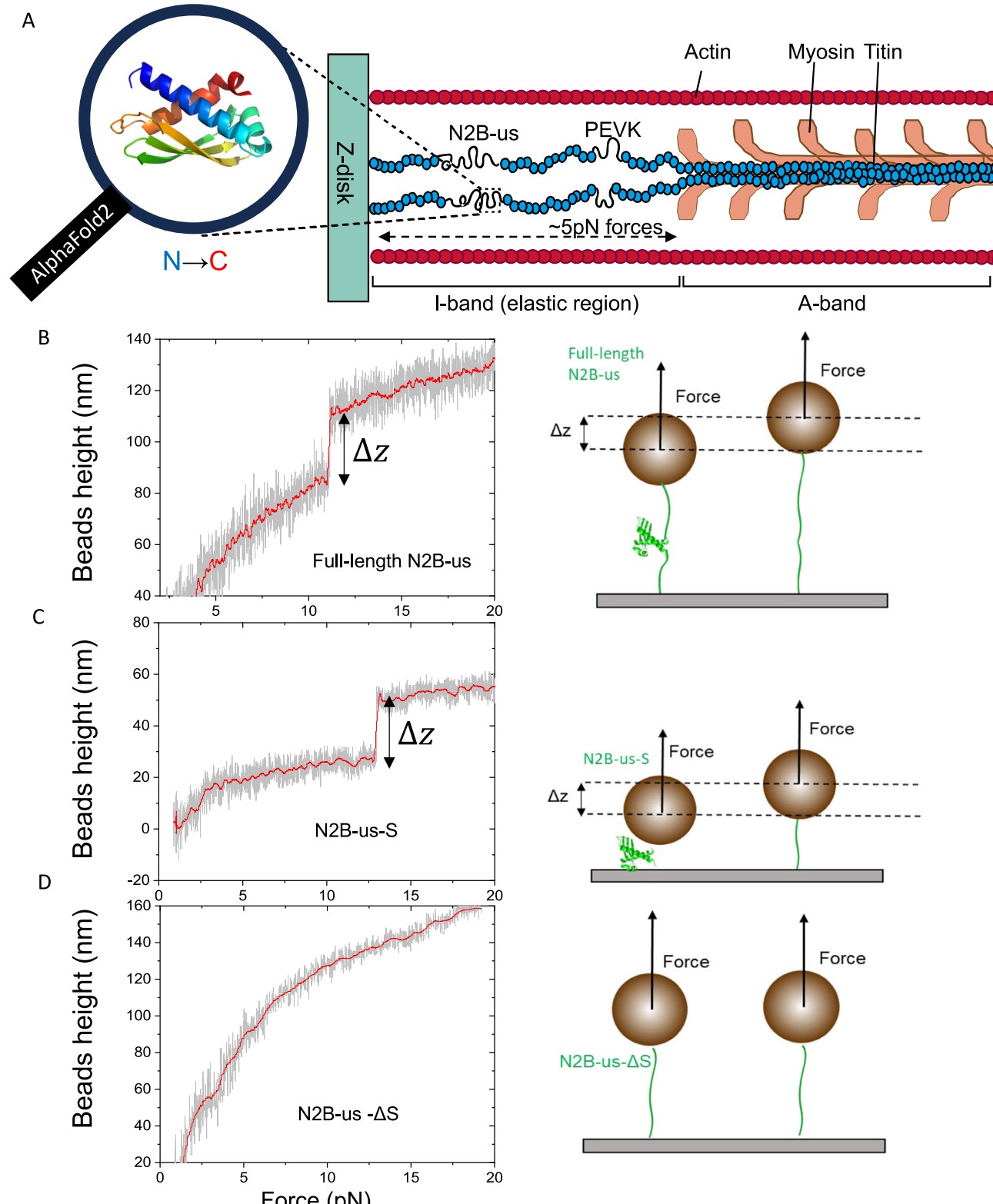

**Fig. 1 | N2B-us structural prediction by Alphafold2 and single-molecule validation. A** Schematic illustration of N2B-us position in sarcomere (right) and N2B-us structural domain (115a.a.) predicted by AlphaFold2 (left). The structure of the 115-amino-acid long domain contains three β-strands sandwiched between two α-helical regions; color indicate N to C terminal. **B** Left panel: representative trace of the force-bead height curve during force-increase scans at a force loading rate of 1pN/s for full-length N2B-us. The unfolding of the structured domain resulted in the stepwise height increases indicated by ΔZ. Right panel: illustration of corresponding single-molecule detection of the conformations of full-length N2B-us, The Illustration shows the protein construct for N2B-us tethered between a superparamagnetic microbead and a coverslip surface, ΔZ indicates the bead height change caused by the structural domain unfolding. **C, D** Same representative traces and illustrations for the predicted structural domain (N2B-us-S), and N2B-us with the deletion of the structural domain (N2B-us-ΔS), respectively. The bead height change was observed for the N2B-us-S construct but not in N2B-us-ΔS. Source data are provided as a Source Data file.

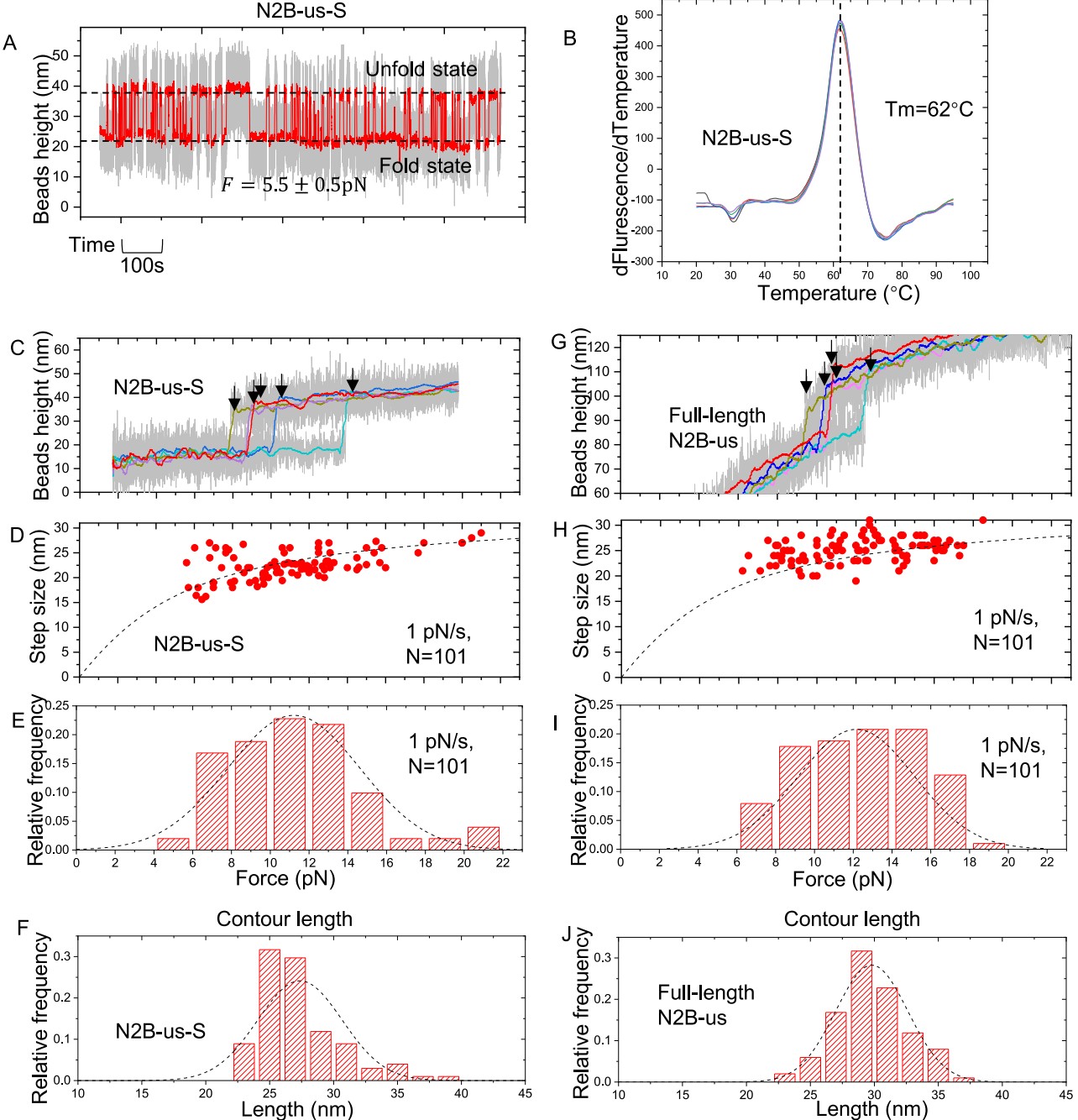

**Fig. 2 | Dynamic transitions between the folded and unfolded states of N2B-us-S.** **A** A representative time trace of the height of a bead attached to a single protein construct containing an N2B-us-S at a constant force of $5.5 \pm 0.6$ pN. The stepwise changes of the bead height indicate spontaneous unfolding and refolding of the N2B-us-S, from which the overall folded-to-unfolded ratio, $\frac{P_{folded}}{P_{unfolded}}$, can be obtained as the ratio of the fractions of time of the two states. **B** Melting curve analysis of the N2B-us structural domain using a protein thermal shift assay. The peak of the curve, observed at 62 °C, signifies the melting temperature of the N2B-us structural domain. **C** Five representative force–height curves of N2B-us-S tether during force-increase scans at a loading rate of 1 pN/s. The colored curves are obtained by 100-point FFT (fast Fourier transformation) smooth of the raw data (gray). The arrows indicate the unfolding events of N2B-us structural domain. **D** Scatter plot depicting domain unfolding step sizes and their corresponding forces. The dashed curve represents the theoretical prediction of these step sizes. **E**, **F** the normalized force and contour length histogram obtained over 101 unfolding events from 5 independent tethers for N2B-us-S. **G–J** Parallel Representation to **C–F** for full-length N2B-us construct, unfolding force, step size, and contour length exhibit similarities to those of the N2B-us-S construct. Source data are provided as a Source Data file.

and $11.1 \pm 1.1$ pN, respectively. The results suggest that the truncation of the N2B-us structural domain from the full-length peptide doesn't significantly affect the mechanical stability of the structure. Based on the data (Fig. 2F, J) and the worm-like chain polymer model for the unfolded state with a bending persistence length of 0.4–0.8 nm. (typical persistence lengths of peptide polymer[36–38]), the contour length of this structural domain is estimated in range of 26.0-30.5 nm

for N2B-S, and 29.5-34.0 nm for the structural domain in the full-length N2B-us, which are comparable to each other.

## FHL2 LIM domain binding sites spread throughout the N2B-us

N2B-us also plays a role in signal transduction by interacting with several signalling proteins including FHL2, a LIM domain protein involved in numerous cellular processes[18,42]. Importantly, dysregulated

FHL2's interaction with N2B-us is believed to be involved in the development of cardiomyopathy[43]. An N2B-us region 79 a.a.–308 a.a. was found to contain FHL2 binding sites[18] using biochemical pull-down assay, but the number of binding sites and their exact positions remain unknown. In addition, it is unclear whether there are additional FHL2 binding sites beyond the previously reported region. To address these questions, we applied AlphaFold2 to predict the structure of the complex of FHL2 and N2B-us and used a magnetic-tweezers-based single-molecule experiment to probe the interactions between N2B-us and FHL2.

AlphaFold2 predicts FHL2's binding to N2B-us with several conformations involving different binding sites in N2B-us fragments (Fig. 3A, more can be found in Figshare Data S1). The model shows a single binding site (peptide of ~5 a.a.) on N2B-us antiparallelly associates with the first zinc finger in each FHL2 LIM domain via hydrogen bonding with the first β-strand in the zinc finger. Figure 3B depicts the predicted aligned error (PAE) plot for the representative complex formed by a full-length N2B-us and a FHL2 molecule. In AlphaFold2, PAE provides a distance error for every pair of residues. This means it offers an estimate of the positional error at residue x when the predicted and true structures are aligned at residue y. The plot was generated by traversing both x and y across all the amino acids in the structure. On the PAE plot, the blue-shaded area represents a highly reliable prediction, corresponding to the binding interface between FHL2 and N2B-us. This suggests a confident prediction of N2B-us binding with FHL2.

To explore additional potential binding sites of FHL2 on N2B-us, we divided N2B-us into five subsegments, each approximately ~100 amino acids long. FHL2 binding sites were predicted within each of these segments using AlphaFold2 (Fig. S10). Additionally, AlphaFold2 predicted FHL2 LIM binding sites within each of the β-strands and the spanning alpha-helical regions of the structural domain (Fig. S11A). In contrast, when the structural domain is folded, no binding was predicted (Fig. S11B), implying that the binding sites for FHL2 in the structural domain are cryptic when the domain is folded. These results suggest that FHL2 binding sites are spread throughout N2B-us, and some of them are cryptic in the folded structural domain. In total, 24 FHL2 LIM domain binding sites were predicted, eight of which are buried in the folded structures (Fig. 3C).

We then tested the extensive distribution of FHL2 LIM domain binding sites throughout N2B-us, as predicted by AlphaFold2, using magnetic-tweezers experiments. Figure 3D shows representative force-bead height curves recorded during a force-increase scan (solid circles) followed by a force-decrease scan (hollow circles), before (red) and after (blue) the introduction of 200 nM FHL2 for full-length N2B-us. The tether was held at each force for 20 s, and the bead height was obtained as the average value over the 20 s. The results suggest a significant FHL2-dependent shift in the force-bead height curves, indicating that FHL2 binding induces a change in the N2B-us conformation.

To probe the binding of FHL2 to the structural domain, we conducted the FHL2 binding assay using the N2B-us-S construct (Fig. 3E). Prior to introducing FHL2, the force-bead height curve exhibited substantial hysteresis, with the force-decrease curve higher than the force-increase curve. This hysteresis has resulted from the unfolding (occurring at higher forces during the force-increase scan) and refolding (occurring at lower forces during the force-decrease scan) of the structural domain. Upon the introduction of 200 nM FHL2, there was a significant up-shift in the force-bead height curves along with disappearance of hysteresis. This outcome suggests the binding of FHL2 also occurs to the structural domain. The increased extension at lower forces likely indicates binding-induced inhibition of refolding.

We also tested the binding of FHL2 to the region outside the structural domain using the N2B-us-ΔS construct. Upon introducing

200 nM FHL2, we observed shifts in the force-bead height curves, indicating binding of FHL2 to this region.

These experiments were conducted repeatedly across five tethers for each construct, as illustrated in Fig. 3G–I. Differences in bead height before and after the addition of FHL2 were measured at three different forces: $1.5 \pm 0.2$ pN, $5.0 \pm 0.5$ pN, and $18.0 \pm 1.8$ pN. These differences were then analysed using a one-sample t-test to determine their statistical significance compared to zero. The results consistently show a decrease in bead height at higher forces and an increase in bead height at lower forces. Specifically, for N2B-us-S, the decrease in the bead height at higher force is less pronounced compared to N2B-us and N2B-us-ΔS, likely due to its smaller size of only 115 amino acids. Additionally, we observed that with an increase in FHL2 concentration, the N2B-us exhibits greater force-bead height curve shifting (Fig. S12).

Together, the results from AlphaFold2 prediction and the single-molecule binding assay show the existence of multiple FHL2 binding sites in both the structural and disordered regions of N2B-us, which may organize N2B-us into different conformations. We also note that the observed binding is specific to FHL2 as we do not observe similar binding for two other LIM domain containing proteins, LIM domain only protein 3 (LMO3), and LIM/homeobox protein2 (LHX2), as shown in Fig. S13.

## Mechanical regulation of full-length N2B-us binding to FHL2

Having demonstrated extensive binding of FHL2 to both the disorder region and the region bearing the structural domain, we proceed to test whether the force has a significant impact on the binding of FHL2 to the full-length N2B-us. We conducted experiments using two distinct force-modulation procedures. In the first procedure (Fig. 4A), a constant force of 1 pN was applied for 10 s. This was immediately followed by a force jump to 8 pN for 20 s and then a return to 1 pN. We measured the difference between the average bead heights during the initial and final 10-second intervals at 1 pN. In the second procedure as a control, a separate tether was subjected to a steady force of 1 pN for 40 s (Fig. 4B). Should FHL2 bind during these procedures, we anticipate observing a non-zero difference in the average extensions.

In the first procedure, we observed a notable average extension difference of approximately 10 nm in the presence of 200 nM FHL2 and a negligible average extension difference when FHL2 was absent (Fig. 4C). This suggests that FHL2 binds to N2B-us during the 20-second application of 8 pN force.

In stark contrast, Fig. 4D shows a negligible average bead-height difference for the second procedure, irrespective of the presence of FHL2, implying that there is no significant binding of FHL2 over the observed time course due to the omission of exposing the protein construct to the higher force of ~8pN.

The force-dependent binding of FHL2 to N2B-us was further validated using a single-molecule fluorescence imaging assay employing a total internal reflection microscope (iLAS2_TIRF). In this assay, N2B-us was subjected to constant force via a magnetic-tweezer setup while being exposed to 100 nM quantum dots (QD, DiagNano™ GSH CdS/ZnS Quantum Dots, 450 nm) conjugated with FHL2. Imaging was conducted with 405 nm laser excitation, producing a 200 nm evanescent field. Stable localized fluorescent signals were observed only when FHL2-QD complexes bound to N2B-us and remained within the evanescent field for a specific duration. To maintain the superparamagnetic bead outside the evanescent field, a 3-kbp DNA handle was introduced as a spacer between the superparamagnetic bead and the N2B-us protein (Fig. 5A). Additionally, to locate the tether beneath the microbead, a 561 nm laser and the RFP channel were employed to image Sytox Orange-dyed DNA handles (Fig. S14). The 576 nm × 576 nm area surrounding the fluorescence signal from the DNA handle underneath the microbead was selected as the region of interest for QD signal imaging.

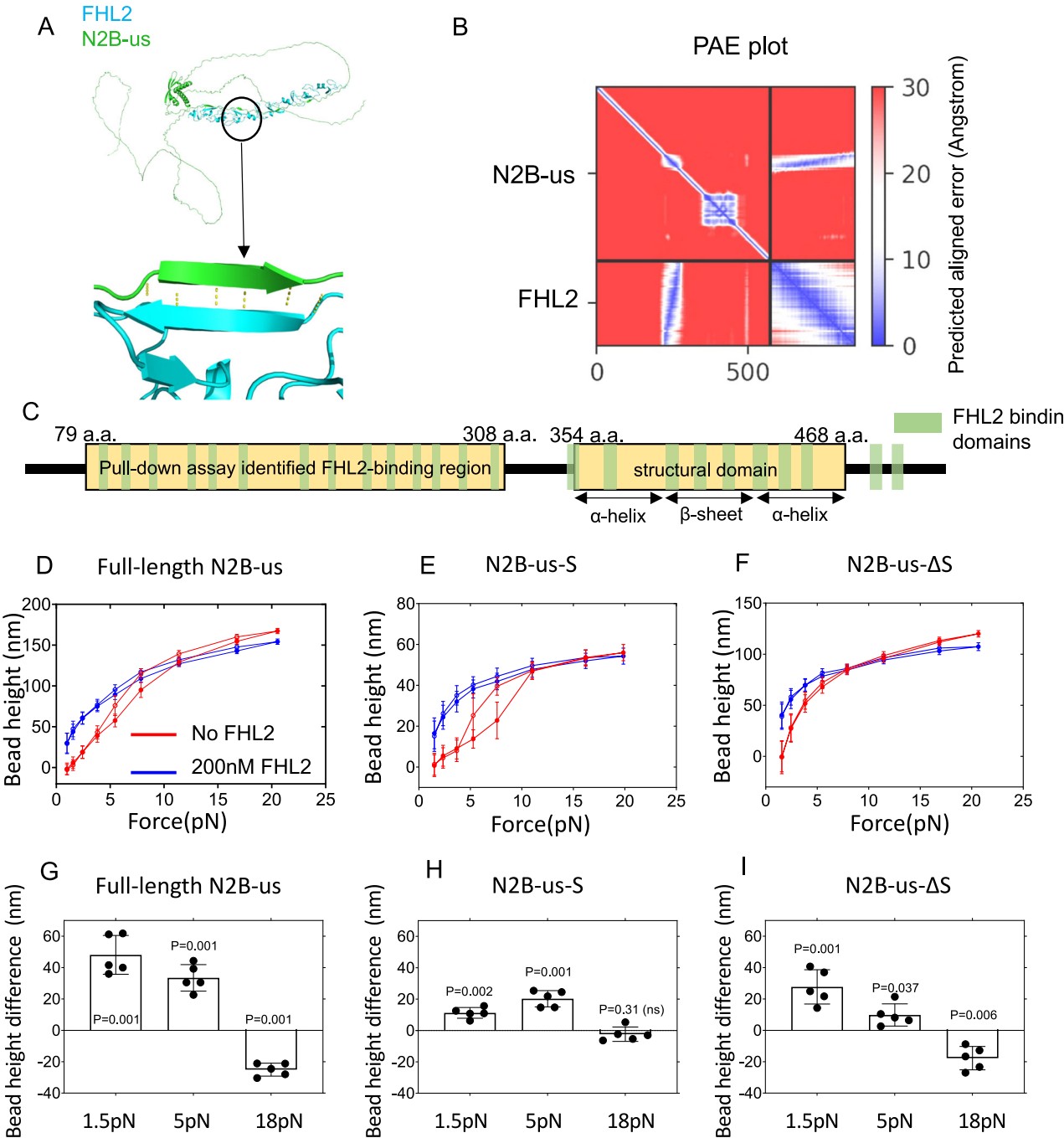

**Fig. 3 | FHL2 binds to multiple sites on N2B-us. A** Structure of full-length N2B-us in complex with FHL2 from AF2. The binding interface shows anti-parallel beta-strand conformation. **B** Predicted aligned error (PAE) plot for the predicted structure. The area corresponding to binding interface shows low error (blue), suggesting that the prediction for the binding interface is confident. **C** FHL2-binding sites summary map for N2B-us. In total 24 FHL2-binding sites were predicted on N2B-us, 14 of which were in the previously reported segment identified by protein pull-down assay (79aa-308aa), 8 of which were cryptic in the structural domain (354aa-468aa). The secondary structures in the N2B-us-S are labelled. **D** Representative force - height curves obtained from a full-length N2B-us tether during force-increase (solid-circles) and force-decrease (open-circles) scans, at each force bead height were measured for 20 s and take average value (4000 data points for each force, the error bars indicate standard deviation) before (red) and after (blue) introduction of 200 nM FHL2. The blue curves are higher than the red curves at forces below 8 pN. This experiment was repeated in five independent tethers. **E** Similar representative force - height curves obtained from a N2B-us-S, similarly higher blues curves than the red curves over pN force range was observed. **F** Similar representative force - height curves obtained from a N2B-us-ΔS tether; bead height increase over pN force range was also observed. **G**–**I** Quantification of bead height difference between before and after addition of FHL2 at tension Levels of 1.5 pN, 5 pN, and 18 pN for three constructs. Results compiled from five independent experiments for each construct. The error bars indicate standard deviation. Two sided one-sample *t*-test was applied to evaluate if the observed differences significantly deviate from 0 for each group. Source data are provided as a Source Data file.

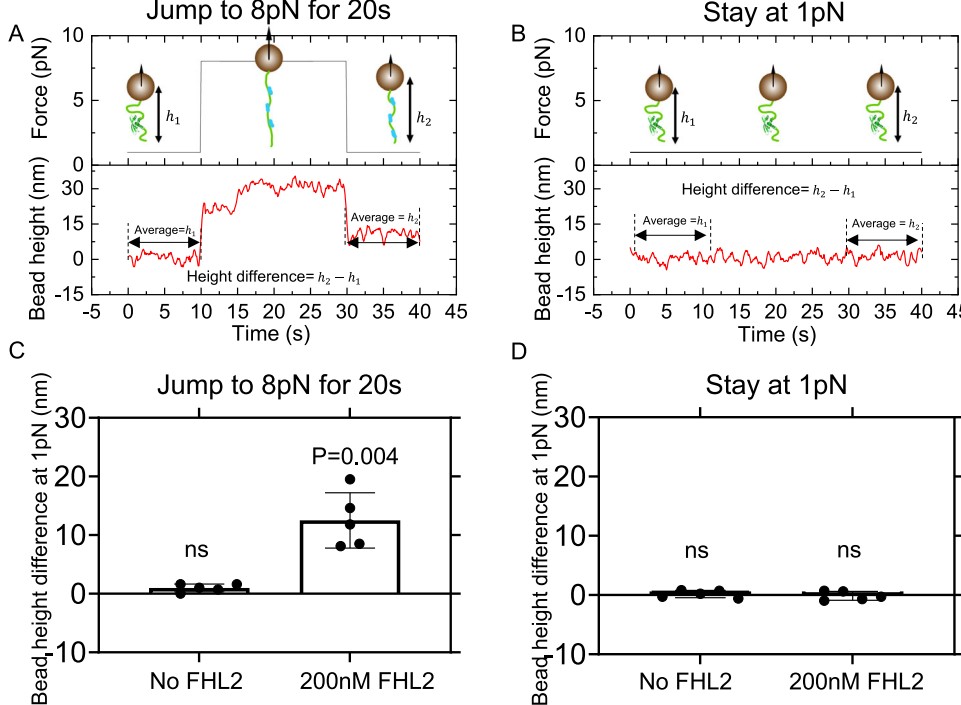

**Fig. 4 | Mechanical regulation of N2B-us binding to FHL2. A, B** Illustration of two different experimental procedures, with the tether in each state also illustrated. The green color indicates N2B-us, and the blue color indicates FHL2. **A** Jump to 8 pN procedure: The bead was initially held at 1 pN for 10 s, then the force was increased to 8 pN for 20 s, and subsequently returned to 1pN. The bead height difference was obtained by measuring before and after this 20-second interval under a 1 pN force. **B** Stay at 1 pN procedure: The force was consistently maintained at 1 pN. The bead height was measured before and after a 20-second interval under a 1 pN force, and the difference was calculated. **C** Bead height difference quantification under jump

to 8 pN procedure: No significant difference in bead height was observed in the absence of FHL2 ($n = 5$, $p = 0.3$). Conversely, with 200 nM FHL2, the bead height difference was significantly greater than 0. **D** Bead height quantification under stay at 1 pN procedure: the bead height remained unchanged regardless of the presence or absence of FHL2 ($n = 5$, $p = 0.57$, $p = 0.71$, respectively). For each procedure, data from five independent tethers were analyzed using a two sided one-sample t-test. The error bars indicate standard deviation. Source data are provided as a Source Data file.

Videos were recorded for each tether under two conditions: forces of 1 pN and 10 pN, with each video frame captured at an exposure time of 50 milliseconds. A total of 1000 frames were captured for each condition and tether. The fluorescent intensity within the tether area, as indicated by DNA handle imaging, was calculated for each frame. The average value for each tether was then normalized against the background value. Image processing was performed using ImageJ without any prior modification of the image intensity before quantification. Figure 5B displays representative traces, Fig. 5C presents data from 19 independent tethers. The results show that under a 10 pN force, the average relative fluorescent intensity is significantly higher than under a 1 pN force. Conversely, in the absence of FHL2, there was no significant difference between the 10 pN and 1 pN conditions (Fig. 5D), suggesting that the observed effect is not due to nonspecific binding attributed to force-induced adhesiveness of N2B-us. These findings indicate that the 10 pN force promotes the binding of FHL2 to the N2B-us region, consistent with observations obtained from the binding-induced extension change (Fig. 4C-D).

Together, the results demonstrate that higher force exposure facilitates binding of FHL2 to the full-length N2B-us, and the binding leads to a longer extension at lower forces of a few pN.

### Mechanical activation of N2B-us structural domain's binding to FHL2

The above results lead to a hypothesis that the higher force facilitates binding via mechanical unfolding of the structural domain that exposes the cryptic binding sites.

To test it, we employed a force-cycle assay. In this assay, a force progressively increased at a rate of 1 pN/s from 1 pN to 25 pN, which

caused mechanical unfolding of the structural domain, then decreased at the same rate back to 1 pN. Prior to the addition of FHL2, distinct unfolding and refolding signals were observable (Fig. 6A, black trace). However, upon introducing 200 nM FHL2, following a force-induced unfolding of N2B-us-S, the protein did not refold as evidenced by the absence of the refolding signal (Fig. 6A, red trace), and the bead height at 1 pN was significantly elevated. In subsequent force cycles, no unfolding signal was detected (Fig. 6A, blue trace), indicating inhibited refolding as a result from FHL2 binding. The mechanical unfolding facilitated binding of FHL2 to N2B-us-S, which in turn inhibits refolding of N2B-us-S at lower forces, was also confirmed at 37°C (Fig. S15).

Alternatively, in the force-cycle assay conducted across a range of 1 pN to 6 pN, during which the unfolding of N2B-us-S is rare, upon the introduction of 200 nM FHL2, there was no discernible shift in the force-height profile compared to N2B-us-S before the introduction of FHL2. Two examples are illustrated in Fig. 6B, with additional data provided in Fig. S16. While at 37°C lower forces cannot induce FHL2 binding neither (Fig. S17). This finding suggests that FHL2 did not spontaneously infiltrate the structure at low force levels where unfolding of the structural domain did not occur.

The experiment was repeated multiple times, and the difference in bead height at 1 pN before and after each force cycle was recorded and quantified in Fig. 6C. In scenarios without FHL2, there were no changes in bead height observed, regardless of whether the force was scanned up to 25 pN or 6 pN (Fig. 6C, right panel). In contrast, in the presence of 200 nM FHL2, a significant increase in bead height was observed when the force was scanned from 1 pN to 25 pN during which the structural domain unfolded, but not when scanned from 1 pN to 6 pN during which the structural domain remained folded (Fig. 6C, left

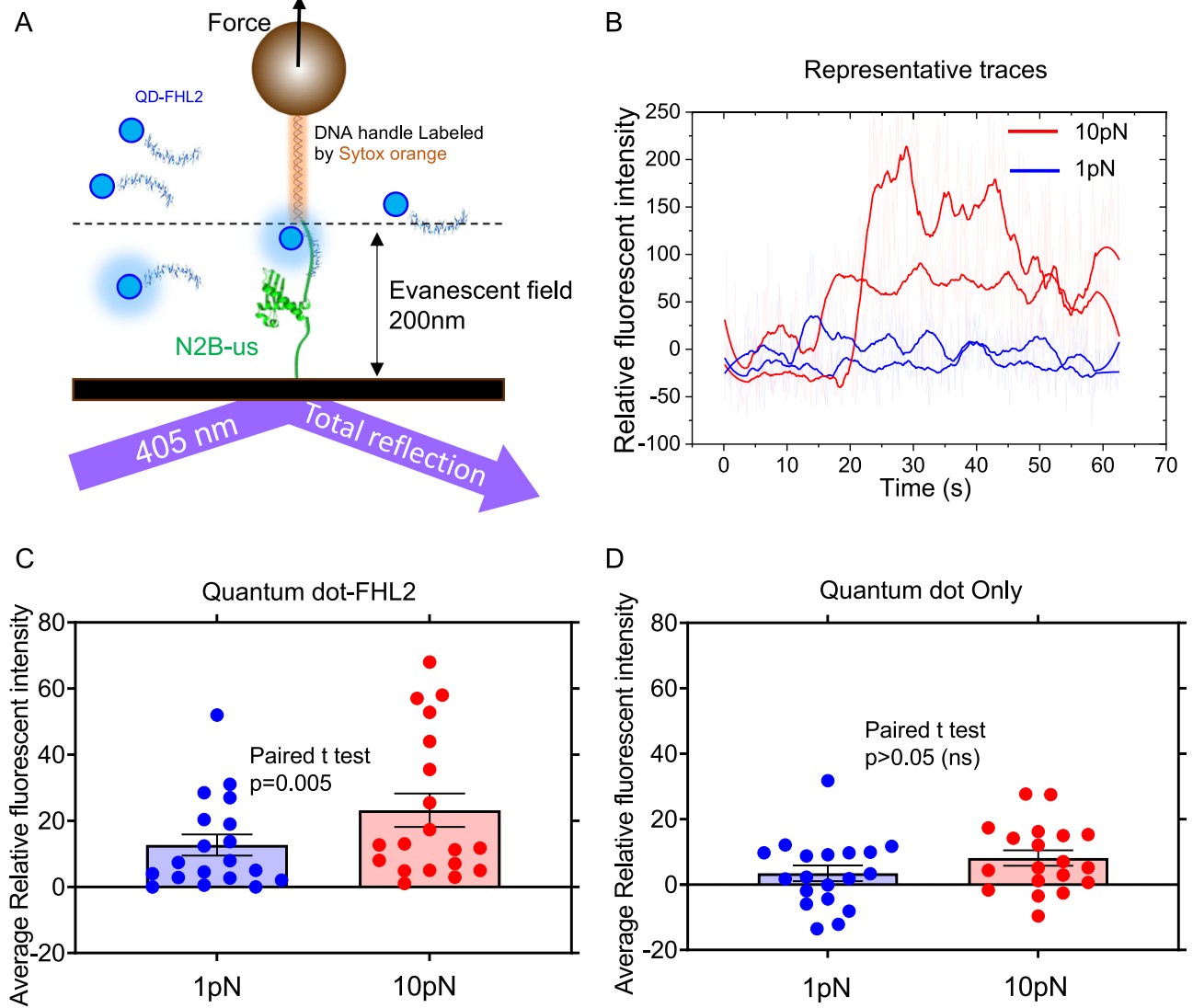

**Fig. 5 | TIRF-magnetic tweezer experiment for observing FHL2 binding to N2B-us under force. A** Setup of the TIRF-magnetic tweezer system. Illustrating the generation of an evanescent field using 405 nm light, with a penetration depth of 200 nm. Quantum dot (QD)-FHL2 that binds to N2B-us remain within the evanescent field, emitting fluorescent light. **B** Representative traces of relative fluorescent intensity under 1 pN (blue) and 10 pN (red) forces. Transparent curves depict raw data, while thicker curves represent FFT-smoothed data with a window of 50 points. Over a duration of tens of seconds, the fluorescent intensity generally remains higher under 10 pN compared to 1 pN. **C** Quantification of the average relative fluorescent intensity under 1 pN and 10 pN forces. Two-tailed paired $t$-test analysis ($n = 19$, $p = 0.005$) demonstrates a significant increase in intensity in the 10 pN group compared to the 1 pN group. **D** Control experiment with only quantum but not FHL2 added to the channel. The average relative fluorescent intensity was not significantly different between 1 pN and 10 pN group (n = 19, p = 0.117), the error bars indicate standard error. Source data are provided as a Source Data file.

panel). The results were subjected to a one-sample t-test to assess their deviation from zero. Figure 6C contains only five data points in the context of FHL2 due to the absence of refolding post the initial FHL2 binding to the unfolded N2B-us-S during the first force cycle. In contrast, there are 20 data points for the control experiments conducted without FHL2.

Collectively, these findings indicate that the binding of FHL2 to the N2B-us structural domain is strongly facilitated by the mechanical unfolding of N2B-us-S. This force-induced unfolding is critical for exposing the LIM domain binding sites essential for interaction with FHL2, as depicted in Fig. 6D. This process elucidates the mechanism by which force regulates N2B-us's binding to FHL2.

## Discussion

In summary, our studies have employed an integrated approach that combines AlphaFold2 structural and binding predictions with single-molecule manipulation to reveal a mechanosensitive structural domain in the N2B-us region, as well as a plethora of FHL2 LIM domain binding sites in N2B-us that leads to mechanically sensitive, high-affinity multivalent binding to FHL2. Figure 7 illustrates the different configurations of the FHL2-N2B-us complex formed at varying forces.

The N2B-us region has conventionally been considered intrinsically disordered. However, analyses employing DISOPRED and PSIPRED have uncovered the presence of secondary structural motifs, such as strands and helices, throughout N2B-us (Fig. S18)[44,45]. Furthermore, a prior single-molecule study noted that N2B-us exhibited a collapsed conformation at low forces[46]. While these preceding studies suggest the presence of secondary structural motifs, it remains uncertain whether a stable tertiary structure may exist and where it might be located within N2B-us. In this study, AlphaFold2 predicts the presence of a well-folded structural domain (~115 amino acids) within N2B-us. This domain was successfully purified and demonstrated to be

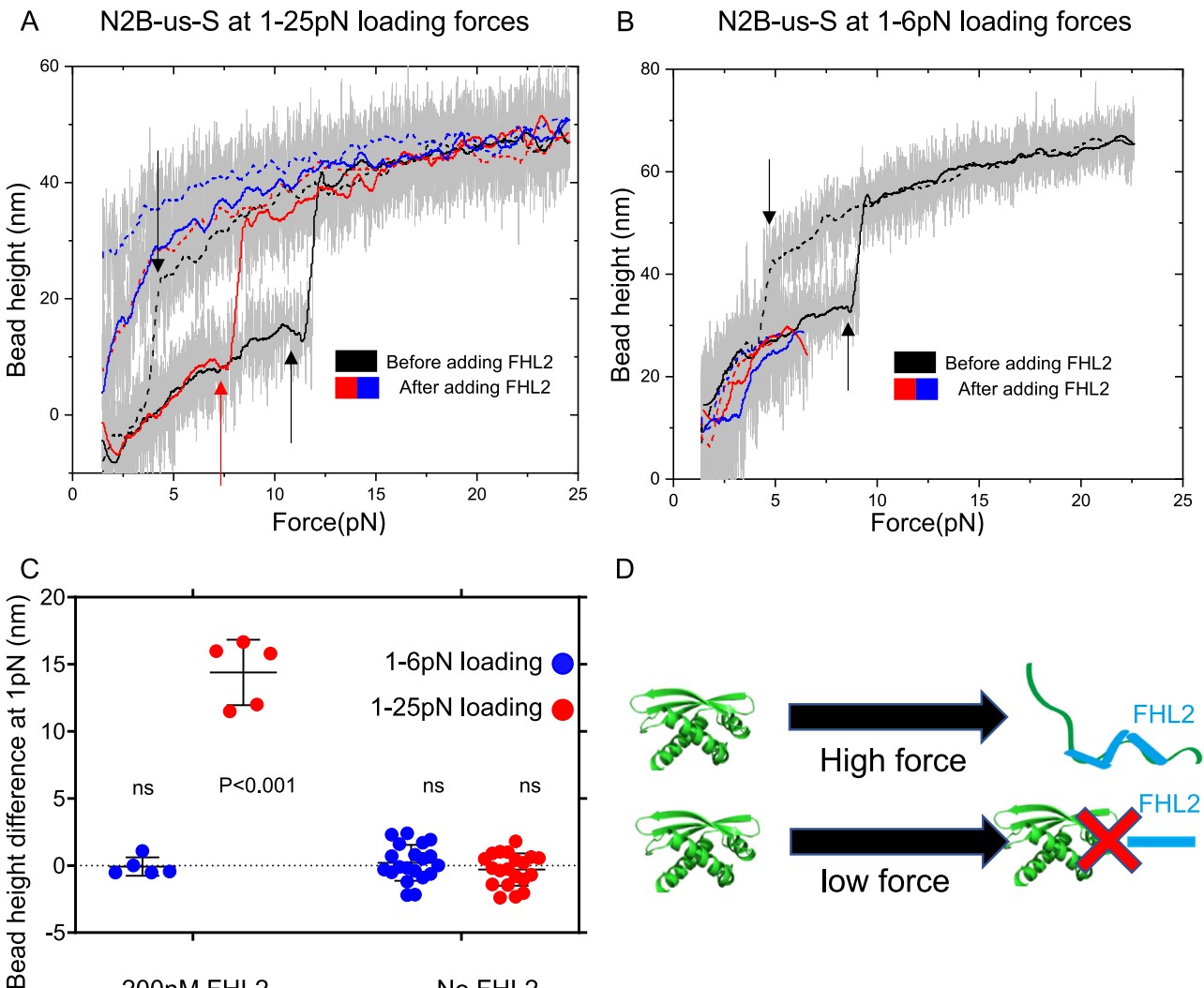

**Fig. 6 | Force-activated N2B-us-S binding to FHL2. A** Force-bead height curve of N2B-us-S during force increase (solid lines) and decrease (dashed lines), before (black) and after (coloured) adding 200 nM FHL2, smoothed by 100-point FFT (grey represents raw data). Initial black arrows denote N2B-us folding/unfolding events. After 200 nM FHL2, the first force increase shows an unfolding event (red arrow) but lacks refolding, leading to a consistently unfolded state in subsequent cycles (blue). The bead height at 1 pN become significantly higher. **B** In a separate experiment, following stretching to 25 pN (showing unfolding/refolding via black arrow), reducing force upper limit to 6 pN after FHL2 addition shows N2B-us remains folded, displaying stable bead height across force changes. **C** Analysis shows low force does not affect bead height with 200 nM FHL2 ($n = 5$, $p = 0.80$), while high force leads to significant increases ($n = 5$, $p = 0.0002$). Absence of FHL2 shows no change at any force level (right panel), validated by two-sided one-sample t-test from five independent data points for 200 nM FHL2 group and 20 data points for no FHL2 group ($n = 20$, $p = 0.50$, $p = 0.27$, respectively). **D** N2B-us-S requires mechanical unfolding for FHL2 binding (upper panel); absence of unfolding prevents binding (lower panel). The error bars indicate standard deviation. The middle lines indicate mean value. Source data are provided as a Source Data file.

thermodynamically stable using protein thermal shift assays and mechanically stable using single-molecule force-dependent unfolding assays.

Our findings suggest that the 572 amino acids of N2B-us consist of a 457 amino acid unstructured peptide region and a 115 amino acid structural domain. The presence of this structural domain is likely conserved in other vertebrates, as homologous sequence domains have been found in other organisms such as mice, sheep, zebrafish, and rats using BLAST with the human N2B-us structural domain sequence. AlphaFold2 predicts that these homologous domains all form highly similar structures containing three beta strands and two helical domains (Fig. S19).

Previous pull-down assay suggested that N2B-us interacts with FHL2 via a region of 79a.a.-308a.a.[18]. However, the exact FHL2 binding sites on N2B-us were not identified. In this work, AlphaFold2 predicts twenty-four FHL2 LIM domain binding sites, each being a linear peptide of 3-5 a.a. Thirteen sites are clustered in a disordered region from 79a.a.-308a.a., and eight are clustered and buried in the folded structure. In each cluster, the FHL2 LIM domain binding sites are linked by short peptides (5-15 a.a.). AlphaFold2 also predicted a highly characteristic structure of the complex formed by an FHL2 LIM domain and a binding site on N2B-us. The existence of many FHL2 LIM domain binding sites throughout the entire N2B-us, including the region bearing the structural domain, is confirmed in our single-molecule assay.

Our single-molecule experiments show that the binding of FHL2 to the disordered region results in slightly increased bead height in pN force range. The shift in the force-bead height curves varies significantly in different experiments, which can be explained by different configurations of FHL2 binding to N2B-us. FHL2 has four and half LIM domains, which can engage different set of binding sites on N2B-us, resulting in various potential FHL2-bound configurations, which may

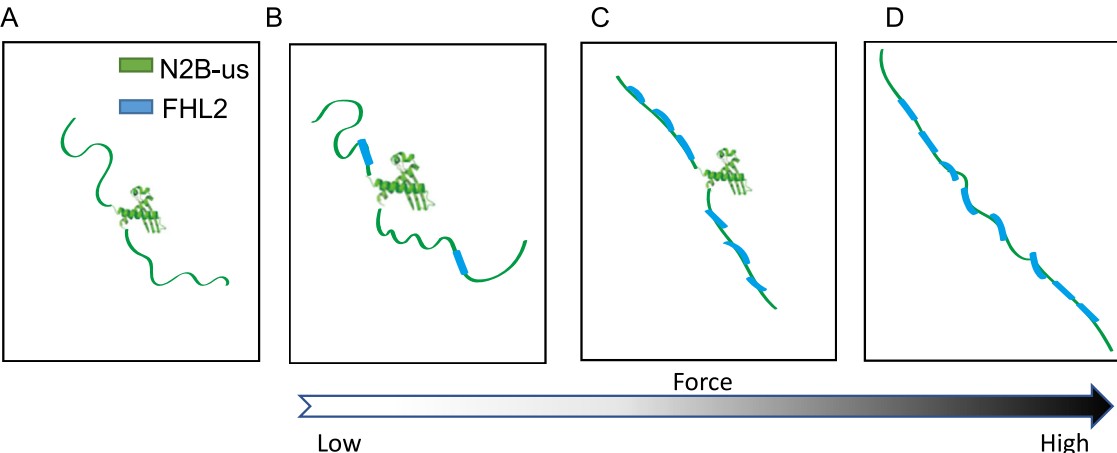

**Fig. 7 | Schematic diagram of force-dependent configurations of the FHL2-N2B-us complexes. A** Unbound N2B-us. **B** FHL2-N2B-us complex formed with FHL2 at low forces where few FHL2 bind to N2B-us disorder region. **C** FHL2-N2B-us complex at medium forces where the more extended unstructured region promotes binding of FHL2 to adjacent FHL2 LIM domain binding sites resulting in stiffening of N2B-us. **D** FHL2-N2B-us complex at high forces where the structural domain is unfolded, and the extended N2B-us promotes binding of FHL2 to adjacent FHL2 LIM domain binding sites resulting in stiffening of N2B-us. These configurations, once formed, can persist during cyclic heart beating.

be associated with different force-dependent extensions (Fig. S20A–C). The binding to the exposed binding sites in unfolded structural domain kinetically inhibits refolding of the structural domain when forces were decreased to below 5 pN. The many FHL2 binding sites make the N2B-us a hot spot for FHL2 binding, which could scaffold downstream binding and potential activation of several important kinases such MAPK and PFK[20].

Our findings suggest that the binding sites located within the structural domain require mechanical unfolding, induced by forces of a few pN, for their exposure. This indicates that FHL2's binding to the structural domain in N2B-us is mechanically activated, a process that can be explained by the large folding energy (approximately −8 kcal/mol) of the structural domain. This high folding energy autoinhibits spontaneous binding in the absence of force.

Moreover, the binding of FHL2 to the 16 predicted FHL2 LIM domain binding sites within the intrinsically disordered region are likely sensitive to mechanical forces. These forces influence the extension of the linkers between adjacent sites, thereby affecting FHL2's binding. The arrangement of clustered FHL2 LIM domains in a linear sequence, with an inter-domain distance of about 3 nm, making it energetically efficient for FHL2 to bind multiple sites on N2B-us when the force-dependent inter-site extension matches the inter-LIM-domain distance. Given the variability in the inter-site lengths, ranging from a few to tens of amino acids, FHL2 will bind to different conformations induced by varying forces. For instance, at higher forces, FHL2 might bind to closely spaced sites stabilizing an extended conformation even as the force decreases, explaining the observed extension increase compared to the unbound N2B-us-ΔS at lower forces in the force scanning assay (Fig. 3F, I). Conversely, at low forces, FHL2 could engage more distal sites along the contour on the N2B-us-ΔS, which becomes closer under lower forces, stabilizing a compact structure which was also observed in our experiments (Fig. S20D).

The results collectively indicate that the configuration of FHL2's LIM domain sites facilitates FHL2 binding to the disordered region across a spectrum of forces, along with further binding to the structural domain, which needs its mechanical unfolding. Crucially, FHL2's interaction with N2B-us is believed to be pivotal in controlling cardiac hypertrophy and atrophy[43]. Thus, these findings point to a mechanical regulation of the N2B-us and FHL2-mediated mechano-transduction pathway.

In the sarcomere, tension on the I-band of titin is primarily regulated by its flexible disordered regions, with a partial contribution from structural domains that may undergo folding and unfolding transitions[47,48]. Typical sarcomere lengths (SL) in human cardiomyocytes range from ~1.7–2.2 μm[49,50]. The thick filament length (TFL) in striated muscle is ~.5–1.6 μm[51,52], and the cardiac muscle exhibits a Z-disc thickness (ZT) of ~100 nm[53]. Therefore, the estimated I-band extension of titin, given by (SL-TFL-ZT)/2, spans a range of roughly 50–300 nm. Taking into account the force-extension curves for all structural domains (41 Ig domains and the 115 amino acid structural domain in N2B-us) along with the approximately 1019 amino acids of disordered regions (PEVK, the disordered N2B-us region, and the linkers between Ig domains, data from UniProt, Q8WZ42), the I-band titin tension can be estimated to be ~1–5 pN, provided all structural domains remain folded (Supplementary note 3). This estimation aligns with previously estimated titin tension derived from mechanical stretching of muscle fibers[15]. Our observation that significant unfolding of the N2B-us structural domain occurs at forces above 4 pN (Fig. S7) falls within this predicted tension range.

This study demonstrates a powerful integration of AlphaFold2 prediction and single-molecule experiments to reveal the structure-interaction relationship of proteins whose structures have not been solved experimentally. Single-molecule mechanical manipulation can provide useful information on the dynamic protein conformations and interactions. Much deeper insights can be obtained if the structural information of the proteins and the protein-protein complexes can be integrated into the interpretation of the results from the single-molecule manipulation studies. Unfortunately, the structures of many proteins have not been solved, and hence the interpretations of single-molecule manipulation studies of such proteins are much less direct and often relies on speculations. This underscores the importance to integrate Alphafold2 structural prediction with single-molecule manipulation studies of force-bearing proteins.

This study marks significant advances by unveiling a previously unidentified structural domain in N2B-us and highlighting its force-dependent binding with FHL2, thereby offering insights into the mechano-sensing functions mediated by N2B-us. Despite these contributions, it is crucial to recognize the study's limitations. The physiological significance of the discovered structural domain remains unclear. This is especially relevant considering that FHL2 binding sites are not confined to the structural domain but also exist in the exposed disordered region. Rather than diminishing the study's impact, this limitation serves as a prompt for future research directions.

## Methods

### AlphaFold2 prediction and PDB file analysis

Protein sequences of Titin N2B-us and FHL2 were obtained from Uniport (Q8WZ42 and Q14192), the sequence for full-length N2B-us is:

DMTDTPCKAKSTPEAPEDFPQTPLKGPAVEALDSEQEIATFVKD
TILKAALITEENQQLSYEHIAKANELSSQLPLGAQELQSILEQDKLTPE
STREFLCINGSIHFQPLKEPSPNLQLQIVQSQKTFSKEGILMPEEPETQAVL
SDTEKIFPSAMSIEQINSLTVEPLKTLLAEPEGNYPQSSIEPPMHSYLTSV
AEEVLSPKEKTVSDTNREQRVTLQKQEAQSALILSQSLAEGHVESLQSPD
VMISQVNYEPLVPSEHSCTEGGKILIESANPLENAGQDSAVRIEEGKSLRFPL
ALEEKQVLLKEEHSDNVVMPPDQIIESKREPVAIKKVQEVQGRDLLSKESL
LSGIPEEQRLNLKIQICRALQAAVASEQPGLFSEWLRNIEKVEVEAVNITQE
PRHIMCMYLVTSAKSVTEEVTIIIEDVDPQMANLKMELRDALCAIIYEEIDIL
TAEGPRIQQGAKTSLQEEMDSFSGSQKVEPITEPEVESKYLISTEEVSYFNV
QSRVKYLDATPVTKGVASAVVSDEKQDESLKPSEEKEESSSESGTEE-
VATVKIQEAEGGLIKEDG.

The sequence for N2B-us structural domain is:

IPEEQRLNLKIQICRALQAAVASEQPGLFSEWLRNIEKVEVEAVNITQE
PRHIMCMYLVTSAKSVTEEVTIIIEDVDPQMANLKMELRDALCAIIYEEI-
DILTAEGPRIQQGAKT.

The sequence for FHL2 is:

MTERFDCHHCNESLFGKKYILREESPYCVVCFETLFANTCEECGKPI
GCDCKDLSYKDRHWHEACFHCSQCRNSLVDKPFAAKEDQLLCTDCYS
NEYSSKCQECKKTIMPGTRKMEYKGSWHETCFICHRCQQPIGTKSFIPKD
NQNFCVPCYEKQHAMQCVQCKKPITTGGVTYREQPWHKECFVCTA
CRKQLSGQRFTARDDFAYCLNCFCDLYAKKCAGCTNPISGLGGTKYISF
EERQWHNDCFNCKKCSLSLVGRGFLTERDDILCP DCGKDI

AlphaFold2 code were ran in google colab page (https://colab.research.google.com/github/sokrypton/ColabFold/blob/main/AlphaFold2.ipynb)[54], configurations setting are as follows: msa-method=mmseqs2, pair_msa=False, max_msa=512:1024, subsample_msa=True, num_relax=0, use_turbo=True, use_ptm=True, rank_by=pLDDT, num_models=5, num_samples=1, num_ensemble=1, max_recycles=3, tol=0, is_training=False, use_templates=False. The exported PDB file were visualized and analyzed in Pymol 2.4.1. Sequence alignment was performed in TCOFFEE[55].

### Protein expression and purification

The pET151-6His-Avi-2I27-N2B-2I27-Spy plasmid was prepared as previously described[46]. Truncation of N2B-us folding domain was obtained by PCR (Q5® High-Fidelity DNA Polymerase, M0491) and Q5® High-Fidelity DNA Polymerase (M0491S) reaction, experiments were performed by the manufacturer's instructions. Deletion of N2B-us folding domain was accomplished by PCR and Dpn1 (NEB, R0176S) reaction with subsequent KLD (NEB KLD Enzyme Mix, M0554S) reaction, procedures were as described in manufacturer's protocol. The sequences of primers, which were from IDT RxnReady® Primer Pools, were listed in table S1. All constructs were validated by full sequencing of the open reading frame (Axil Scientific) and expressed in E. coli BL21 strain and purified using His-tag Coolum.

N2B-us structural domain protein for protein thermal shift assay was prepared separately. gBlock DNA fragment was ordered from IDT, the sequence is attached in table S2. The DNA fragment was ligated to pGEX vector using NEBuilder HiFi DNA Assembly. The primers used to linearize pGEX vector is in table S1. A thrombin cutting sequence was added between the GST tag and the target protein of interest. The construct was expressed in E. coli BL21 strain and purified using GST binding Coolum then cut by thrombin protease. The N2B structural protein was collected with gel filtration chromatography while the GST tag was removed. All protein purity was evaluated by SDS-polyacrylamide gel electrophoresis, and protein concentration was measured by absorbance at 280 nm.

Recombinant GST-FHL2 protein was purchased from Abnova (H00002274-P01), which was expressed in Wheat Germ.

### Single-molecule manipulation

Single-molecule N2B-us stretching experiments were performed on homemade magnetic-tweezers. The channels used in the experiments were prepared using the following procedure: First, $20 \times 32$ mm$^2$ coverslips were meticulously cleaned through a sequence of steps. They were subjected to ultrasonication in detergent, followed by acetone, and then distilled water, each for a duration of 20 minutes. Subsequently, the coverslips were completely dried using an oven, and they underwent a 15-minute treatment with oxygen plasma. To prevent disturbance when changing buffer in the channel, we add a microwell array to the coverslips as described in[56], then covered by $18 \times 18$ mm$^2$ clean coverslip (Deckglaser) with two parallel stripes. The laminar flow channel was further treated with 1% glutaraldehyde for 1 h. After fully rinsed by PBS (Phosphate Buffered Saline, Research instruments, SH30256.01), 3 µm diameter amine-coated polystyrene beads were added to the channel and incubate for 20 min. Spy catcher expressed in E coli BL21 and purified by His tag column. and incubate in a concentration of 200ug/ml overnight. Afterward, 1% BSA in PBS buffer was added to the channel and incubate overnight for blocking.

Spy[29] and avi[28] tagged N2B-us protein were diluted to 0.002 mg/ml and were introduced in the channel and incubated for 30 min before the introduction of 2.8 µm diameter streptavidin-coated paramagnetic beads (M270 streptavidin, Dynabeads), incubate for 20 min for forming tethers. The buffer for the single-molecule manipulation experiment is PBS with 1% BSA and 10 mM DTT. During experiments, the force on each bead was determined based on their thermal fluctuations under force, with a relatively small error (-10%).

Additionally, we employed a temperature control system to examine protein behaviour at physiologically relevant temperatures. We attached a heating film to the channel, a power supply (TENMA 72-2690) regulated the voltage applied to the heating film, and the temperature was calibrated using a needle thermocouple (Thermometer TASI-600). Further details can be found in reference[57].

Raw extension data were recorded at a sampling rate of 200 Hz, data visualization and analysis were performed in OriginPro 2019b. To improve the clarity of data representation, some figures were smoothed using the fast Fourier transform (FFT)-smooth function in OriginPro 2019b. The force-extension curves were obtained by staying in each force for 20 s and taking the average extension, data processing was performed by in-house written Python 3.8 code and ran in spyder (anaconda 3). The illustration was performed in GraphPad Prism 9.

### Protein thermal shift assay

The protein thermal shift assay was performed using GloMelt™ Thermal Shift Protein Stability Kit. N2B-us structural domain protein concentration used is around 1 mg/ml. Procedures were as described in manufacturer's protocol. The assays were performed on Bio-Rad CFX 96 Real-Time PCR Detection System. The temperature increasing rate is 0.5° per min. Five replicates were done in the experiments. The derivative of fluorescence to temperature curves were obtained, and the melting point of the protein was determined by the peak of the curve.

### TIRF-Magnetic tweezer single molecule fluorescence experiment

We enhanced the iLAS2_TIRF system by integrating a pre-calibrated magnet, meticulously controlled by an MP-285 motor to apply force to M270 magnetic beads. The channel treatment process and the working buffer remained consistent with the protocol detailed in the Single-molecule manipulation section.

Moreover, we introduced a 3-kbp DNA handle to act as a spacer between the magnetic bead and the surface. The synthesis of the DNA handle involved ordering two chemically modified primers from IDT (Table S1), with dual biotin modification at the 5' end of the F primer and thiol modification at the 5' end of the R primer. The DNA handle

was obtained through PCR using the aforementioned primers on lambda DNA and purified by PureLink™ PCR Purification Kit (Invitrogen). Subsequently, the DNA handle was covalently attached to the M270 beads (Dynabeads M270-epoxy) via a thiol-epoxy reaction with the epoxy surface. The opposite end of the DNA handle was linked to a full-length N2B-us AVI tag via neutravidin.

In preparation for the experiment, 10 μM FHL2 was pre-incubated with an equal concentration of GSH-coated quantum dots with an emission peak of 450 nm (DiagNano™, GSH CdS/ZnS quantum dots) for 20 minutes at room temperature before centrifugation at 200 g for 5 min to remove the supernatant. The final concentration of quantum dots in the system was 100 nM. To facilitate the marking of the N2B-us location under the bead, 50 nM Sytox Orange DNA dye (Thermal, S11368) was introduced during the experiment to image the DNA handle. Additionally, to prevent quantum dot blinking, 10 mM DTT was added[58].

Imaging was conducted using 405 nm laser excitation, generating a 200 nm evanescent field. Videos were captured under 1 pN and 10 pN conditions for each tether, with each frame having an exposure time of 50 ms, 1000 frames were taken for each tether each condition. The interval between frames is ~0.12 s, thus the total time recorded for each condition is ~2 min. The fluorescent intensity within the tether area was calculated for each frame, and the average value for each tether was normalized by the corresponding average background value. The imaging processing was performed using imageJ without any modification of the image intensity before quantifying. The results were assessed using a paired t-test.

### Reporting summary

Further information on research design is available in the Nature Portfolio Reporting Summary linked to this article.

## Data availability

All data is available at main text or supplementary information and the pdb files from AlphaFold2 are in Figshare Data S1. The Figshare Data S1 has been deposited in the Figshare database under accession code https://doi.org/10.6084/m9.figshare.25745091. Source data are provided with this paper[59]. Source data are provided with this paper.

## Code availability

All the code used in the manuscript has been uploaded to Zenodo with [https://doi.org/10.5281/zenodo.11108385][60].

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

## Acknowledgements

This research is supported by the Singapore Ministry of Education Academic Research Fund Tier 3 (MOE Grant No: MOET32021-0003) and Tier 2 (MOE Grant Nos: MOE-T2EP50123-0008), National Research Foundation, Prime Minister's Office, the Mid-Sized Grant (NRF-MSG-2023-0001), and Ministry of Education under the Research Centers of Excellence program through the Mechanobiology Institute at National University of Singapore (A-0003467-01-00 and A-0003467-00-00). We also would like to thank Dr. Rong Li (National University of Singapore), Dr Pakorn Tony Kanchanawong (National University of Singapore) for encouraging discussions. Also, we would like to thank Dr. Hongying Chen from MBI Protein Expression Core facility for her kind help with protein expression.

## Author contributions

Y.S. performed the experiments; Y.S. and X.L. developed the TIRF-magnetic tweezer system. Y.S. and J.Y. designed the experiments and interpreted the data. W.H. and S.L. contributed to experiments and design of the experimental construct. Y.S. and J.Y. wrote the paper. J.Y. supervised the research. All authors have given approval to the final version of the manuscript.

## Competing interests

The authors declare no competing interests.
