## [Peer Review File · Nature Communications]

Reviewers' Comments:

Reviewer #1:

Remarks to the Author:

In this paper, the authors use single-molecule manipulation by magnetic tweezers and AlphaFold2 predictions for structural and binding properties of a mechanically active region in cardiac titin, the N2B unique sequence (N2Bus). They confirm and extend their previous finding that regions in the N2Bus domain undergo unfolding-refolding transitions at low forces (between ~ 5 and ~ 15 pN) and identify an alpha/beta structural domain of ~ 115 AA which they characterize mechanically. They also confirm and extend the properties of N2Bus as a binding partner of Four-and-a-half-LIM domain protein 2 (FHL2), demonstrating that N2Bus has multiple binding sites for this protein. Binding affinity is increased by stretching N2Bus.

While the mechanical experiments using magnetic tweezers appear to be of high technical quality, several major issues greatly dampen my enthusiasm for this paper:

1. The authors' premises that the titin N2Bus has been considered an intrinsically disordered region is not really true in vivo: Several papers have shown that N2Bus is intrinsically structured by disulfide bonding. This has been reported by Grutzner et al, (*Biophys J* 97:825-34, 2009) and two recent papers using mass spectrometry to detect titin oxidation have demonstrated that N2Bus is oxidized in living heart muscle (Loescher et al, *PNAS* 117:24545-24556, 2020; Herrero-Galán et al, *Redox Biol.* 52:102306, 2022), likely reflecting the formation of disulfide bonds which cross-link this region and limit its extensibility. By contrast, the authors use 10 mM DTT to prevent any oxidation in their in vitro experiments, which raises the question of how physiologically relevant their findings are.

2. The unfolding-refolding transitions in N2Bus reported for the structural domain occur in a force range that is the same as that within which titin Ig domains unfold-refold (Eckels et al, *Annu Rev Physiol* 80:327-351, 2018). Because the elastic titin segment contains dozens of potentially unfolding-refolding Ig domains but only one N2Bus structural domain, it remains to be shown whether unfolding-refolding of the N2Bus structural domain occurs in muscle sarcomeres, in parallel with Ig-domain unfolding-refolding (which has been demonstrated in myofibrils by Rivas-Pardo et al (*Cell Rep* 14:1339-1347, 2016)). At a minimum, the authors should develop a model which includes the mechanical properties of both the N2Bus and the 50+ I-band Ig-domains, to predict whether or not the unfolding of N2Bus may be functionally relevant in real sarcomeres with a full-length titin spring segment.

3. The proposed "mechanosensing" function of N2Bus caused by its stretch-dependent interaction with FHL2 is not convincingly shown. What the authors show is that more FHL2 binds to N2Bus (both the unfolded structural domain and other regions) when this segment is stretched. However, it is not clear how specific this effect is, because no control experiments are performed, e.g., with a protein that does not bind to titin but has a similar size as FHL2. Because N2Bus gets more sticky on stretching, due to the exposure of previously less accessible hydrophobic sites, the increased binding of FHL2 is not evidence of a relevant functional effect. Moreover, the authors do not really address a "mechanosensing" function of N2Bus, as they do not investigate the (potential) downstream effects of FHL2 binding-unbinding to N2Bus in a cardiac muscle cell.

Taken together, new insight into the physiological function of the FHL2-binding to stretched N2Bus titin is not provided.

Minor

1. The manuscript text would need substantial editing.
2. (no page numbers are provided) Intro: The N2B element does not contain tandem-Igs, as stated wrongly. The proximal, middle and distal Ig-regions of titin do contain tandem-Ig segments, but not N2B.
3. As stated above, the specificity of the FHL2 binding to N2Bus is not shown. Can the site(s)

required for N2Bus-binding be identified within the FHL2 molecule? If these sites were mutated, would the reported effects be gone? The authors should use "sham" protein in control experiments (see major issues). They should also systematically use at least two different concentrations of FHL2, to look for a concentration-dependent effect.

4. The sequence of N2Bus is quite variable between species. Is the structural domain well preserved among humans and other vertebrates? Evolutionary preservation can be evidence of a preserved physiological function.

Reviewer #2:

Remarks to the Author:

This manuscript provides an exciting case study of how AlphaFold predictions can be used together with experimentation to gain mechanistic insight into how certain proteins function. Running AlphaFold on an intrinsically disordered region: N2B-us, the authors noticed that part of this region had a well predicted structural domain. Furthermore, additional regions were predicted to bind FHL2. These were experimentally confirmed using stretching experiments.

I don't have any major concerns and find this manuscript to be an exciting demonstration of how AlphaFold predictions can be used to generate testable hypotheses. But I do have a few minor concerns that can be easily addressed.

1) the methods (page 4) specify the uniprot IDs of the proteins (Q8WZ42 and Q14192) but the exact regions used as input to AlphaFold2 are not specified. On page 6, it's indicated that 572aa N2B-us sequence is used. Here it would be good to know what range was used (start and end positions).

2) The authors claim that N2B-us is commonly considered intrinsically disordered (introduction, page 2) and cite a reference 14, but this reference (from 1999) seems rather old. I would suggest running N2B-us region through DISOPRED and PSIPRED (see: <http://bioinf.cs.ucl.ac.uk/psipred>) to see if this region is predicted as disordered and lacking of secondary structure. It would be interesting to see if AlphaFold was actually needed to make this prediction, or if it's something that could've been done with a more traditional method that looks at local conservation patterns.

3) Page 11 - "AlphaFold2 predicts FHL2 binding..." This is where PAE plots should be shown. PAE (predicted aligned error) predicts confidence of interaction between every pair of positions. Just because the final structures are close in 3D space does not necessarily mean that AF2 confidently "predicts" them as binding.

4) It would be interesting to see if the new version of AlphaFold2, specifically trained for predicting protein-protein interactions (AlphaFold-multimer), will also lead to the same observations between N2B-us and FHL2. This can be accessed here:

<https://colab.research.google.com/github/sokrypton/ColabFold/blob/main/AlphaFold2.ipynb>

5) Page 20 - "has precisely predicted more than 98% protein in human genome". I don't think this statement is correct. According to AlphaFold abstract: "almost the entire human proteome (98.5% of human proteins). The resulting dataset covers 58% of residues with a confident prediction, of which a subset (36% of all residues) have very high confidence" - <https://www.nature.com/articles/s41586-021-03828-1>

6) Figure S1 - Why only four of the 5 models are shown? It would be good to know which model is which, and their predicted confidence (plddt, ptm). Maybe color by plddt, as this would be a good indicator that the ordered region is of high confidence.

7) Figure S2 - when describing "binding" the pae plots will be most informative, as these will provide how confident AlphaFold is about the interfaces observed.

8) I would suggest saving all the predicted outputs (all files) from AlphaFold and providing this an

archive for readers. (If this wasn't already done?).

Reviewer #3:

Remarks to the Author:

In the paper entitled "Mechanical regulation of Titin N2B-us conformation and its binding to FHL2", Sun and co-authors use a combination of computational and single molecule magnetic tweezers approaches to identify a mechanically stable region of the N2B-us domain of titin. Additional experiments suggest that this region of the protein is a binding hub for the FHL2 protein, and the binding interaction may serve as a mechanism to regulate protein mechanical stability and also the subsequent protein folding in the absence of force.

The N2B region is generally thought to be an unstructured and elastic region of titin, and hence an important factor governing the passive elasticity of the entire titin protein. Therefore, these results demonstrating a structured and mechanically stable region within the N2B protein is very exciting. Despite this exciting finding, I believe that the data shown in the manuscript is predominantly 'representative traces', with very little analysis to confirm that these traces are indeed representative of a greater number of experiments. In fact, figures 1, 3 and 4 are showing representative traces with no quantification or analysis of the single molecule data. Figure 2 is the only figure with quantification, but panels B, C and D are not mentioned in the text of the manuscript. Furthermore, the authors do not uncover a mechanism explaining exactly what the FHL2 binding does to the structured region of the N2B-us. In the current form, I believe that the data presented in this manuscript is too preliminary for publication in Nature Communications.

Specific comments:

1. In the manuscript the authors use the force spectroscopy hopping curves at 5.5pN to infer that the protein has thermal stability. Recent work (J. Chem. Phys. 151, 185105 (2019); doi: 10.1063/1.5126071) has shown that it is not so simple to extrapolate thermodynamic information from these mechanical measurements. Therefore, if the authors want to state that this region of the N2B is also thermally stable, I would encourage them to do the experimental measurements. Or avoid merging the concepts of thermal and mechanical stability.
2. In figure 1, there is no quantification of the three different constructs, merely a single trace. For example, is the unfolding force of the truncated form the same as the full length? Is the contour length of the truncated vs. full length protein the same? All constructs need to be fully characterized.
3. In figure 2, only panel A is mentioned in the text, the other results are not mentioned. Furthermore, there are details missing in the caption. For example, what is the fitting curve in panel c? Furthermore, figure 2 is the quantification of a trace shown in figure 1 and I believe that these figures can be combined into 1.
4. In the text, the authors state that there are 23 binding sites for FHL2 LIM domains, but in the scheme of binding sites in figure3, I count 24 binding sites represented.
5. It is not clear to me exactly what is shown in the force – bead height curves in figure3. Are these a single protein before and after the addition of FHL2? Or this is the combined data of many measurements? From the text I have understood that each curve is a single protein. If so, I believe the authors should not show a single protein and should show the analysis from many different experiments to show consistent changes in behaviour upon the addition of FHL2. Are the differences statistically significant with multiple molecules / multiple experiments? There should be controls with the addition of something that does not bind to N2B – ideally a mutated form of FHL2 that has no binding affinity to the N2B domain.
6. In figure3, panel E, the N2B-us- Δ S shows hysteresis without FHL2. But in the supplementary figures, it is the N2B-us-S construct that shows hysteresis. Why are there these differences?
7. It is not clear exactly what effect FHL2 has on the mechanical stability and folding behaviour of the N2B region. Are there different mechanisms depending on where FHL2 binds?
8. In figure4 the authors say that they have an upper limit of force of 6pN to not induce rapid unfolding of the protein upon addition of FHL2. I would suggest that if this is the case, that the authors do not perform force-ramp experiments, and instead perform these experiments in force clamp conditions. If the authors held the protein at the hopping force and see an equal distribution

between the unfolded and folded state, then upon addition of FHL2, if there is a change in the protein conformation this will alter the probability of the protein residing in one of the states. These experiments would be much more informative than these incomplete force-ramp protocols.

List of Main Changes and Point-to-Point Responses

A. List of Main Changes

(All the Main changes listed as well as other minor changes in the main text are highlighted in red color.)

Main Changes in Main Text:

- #1. Page 1, Abstract line 4: several words “without stable tertiary structures” are added to clarify the definition of disorder region, in response to the comment 2 From reviewer 2.
- #2. Page 4 Paragraph 2: The amino acids sequence of N2B-us was added, in response to the comment 1 From reviewer 2.
- #3. Page 7, Paragraph 2, the method of “Protein thermal shift assay” is added.
- #4. Pages 8-9, paragraph 2, from line 15. Several sentences “Several papers have shown that N2B us contains six cysteines...” is added to discuss the possible disulfide bonds formed in the N2B-us region as requested by reviewer, in response to the comment 1 From reviewer 1.
- #5. Page 10, paragraph 1, line 12-18. Several sentences “We note that *G0* is estimated...” to discuss the thermal stability of the N2B-us structural domain.
- #6. Page 10, paragraph 2 line 1-6. Several sentences “Nonetheless, the data from the single-molecule...” are added to double confirm the existence of a stable structural domain in the N2B-us based on the quantification of its thermal stability using thermal shift assay, in response to the comment 1 From reviewer 3.
- #7. Page 11, paragraph 1, lines 2-14: Several sentences “We recorded the unfolding forces and step sizes...” are added to show the characterization of full-length N2B-us and compare the result with that from N2B-us-S, in response to the comment 2 From reviewer 3.
- #8. Page 13, paragraph 2, lines 6-11: Sentence “Figure 3B shows the predicted aligned error...” was added to describe the PAE plot in Figure 3B, in response to the comment 3 From reviewer 2.
- #9. Page 16, paragraph 2: A new paragraph is added to describe Figure 3 H-J, which show results of multiple experiments on the force-bead height curves of N2B constructs in response to the binding with FHL2, in response to the comment 5 from reviewer 3.
- #10. Page 16, paragraph 3, lines 6-7: A few words “The higher the FHL2...” are added to state that the higher the concentration of FHL2, the more extended the disordered region of the N2B-us, in response to the comment minor 3 from reviewer 1.
- #11. Page 16, paragraph 4: A new paragraph “We note that the observed binding of FHL2 to...” is added to describe new control experiments that demonstrate the specificity of FHL2 binding to N2B-us, in response to the comment 3 From reviewer 1.
- #12. Pages 18-19: A new section “N2B-us structural domain unfold and FHL2 binding can happen under physiological conditions” is added to demonstrate that the mechanical unfolding of the N2B-us structural domain unfolding and subsequent binding by FHL2 can happen in physiological temperature and force ranges.

- #13. Page 20, paragraph 2, lines 2-4: One sentence “However, analyses using DISOPRED...” is added to describe the prediction result from DISOPRED and PSIPRED, in response to the comment 2 From reviewer 2.
- #14. Page 21, paragraph 1, lines 1-5: A sentence “The presence of this structural domain is likely conserved in other vertebrates, as homologous sequence domains...” is added to show the result of N2B-us structural domain prediction in other organisms, in response to the comment minor 4 From reviewer 1.
- #15. Page 23, paragraph 1: A new paragraph “In sarcomere, the tension transmitted on the I-band of titin...” is added to discuss the force-dependent unfolding and refolding of the N2B-us structural domain in physiologically relevant force and temperature conditions, in response to the comment 2 From reviewer 1.
- #16. Page 23 paragraph 2: A new paragraph “One proposed functional role...” is added to discuss the possible effect of N2B-us structural domain on the force-buffering role of titin.

Main changes in Figures and captions:

- #17. Figure 2:
Panel B is added to show the curve of protein thermal shift assay.
Panel F is added to show the contour length distribution of the unfolded N2B-us-S.
Panels G-J are added to show the characterization of the unfolding steps in full-length N2B-us.
- #18. Figure 3: Panel A is replaced a new figure to show a FHL2/N2B-us complex predicted by AlphaFold2 multimer. Panel B is added to show the PAE plot of panel A. Panels H-J are added to show results of multiple experiments on force-bead height curves of N2B constructs in response to the binding with FHL2.
- #19. Figure 5:
A new figure added to show the result of N2B-us unfolding and binding to FHL2 under physiological force and temperature conditions.

Main Changes in References:

- #20. The following new references are added:
- 34 Grützner, A. et al. Modulation of titin-based stiffness by disulfide bonding in the cardiac titin N2-B unique sequence. *Biophys J* 97, 825-834 (2009).
- 35 Loescher, C. M. et al. Regulation of titin-based cardiac stiffness by unfolded domain oxidation (UnDOx). *Proceedings of the National Academy of Sciences* 117, 24545-24556 (2020).
- 36 Herrero-Galán, E. et al. Basal oxidation of conserved cysteines modulates cardiac titin stiffness and dynamics. *Redox biology* 52, 102306 (2022).
- 41 Tapia-Rojo, R., Mazo, J. J. & Faló, F. Thermal versus mechanical unfolding in a model protein. *The Journal of Chemical Physics* 151, 185105 (2019).
- 45 Jones, D. T. & Cozzetto, D. DISOPRED3: precise disordered region predictions with annotated protein-binding activity. *Bioinformatics* 31, 857-863 (2015).

- 46 Buchan, D. W. & Jones, D. T. The PSIPRED protein analysis workbench: 20 years on. *Nucleic Acids Res* 47, W402-W407 (2019).
- 50 Eckels, E. C., Tapia-Rojo, R., Rivas-Pardo, J. A. & Fernández, J. M. The work of titin protein folding as a major driver in muscle contraction. *Annual review of physiology* 80, 327 (2018).
- 51 Rivas-Pardo, J. A. et al. Work done by titin protein folding assists muscle contraction. *Cell Rep* 14, 1339-1347 (2016).

Main Changes in Supplementary Information:

- #21. Figure S1 PAE plot of each prediction were added to the figure. The protein structure models were colored by pLDDT. pLDDT plot was added.
- #22. Figure S2-Figure S4 additional representative traces of N2B-S, full length N2B-us and N2B- ΔS , respectively.
- #23. Figure S5 New figure about full length N2B-us stretching without DTT
- #24. Figure S7 New figure of protein thermal shift assay for goat IgG as control.
- #25. Figure S8 PAE plot added to the right panel of FHL2 binding to truncated N2B-us fragments.
- #26. Figure S9 PAE plot added to the right panel of FHL2 binding to N2B-us structural domain motifs.
- #27. Figure S11 new figure of N2B-us force-bead height curve under different FHL2 concentrations.
- #28. Figure S12 new figure of N2B-us force-bead height curve with addition of LMO3 and LHX2 protein as control.
- #29. Figure S14 new figure, additional traces of N2B-us-S under 4-4.8pN force in physiological temperature.
- #30. Figure S15 new figure, additional traces of adding FHL2 to N2B-us-S under 4.5pN constant force.
- #31. Figure S16 new figure, traces of N2B-us-S under 1.5, 2.5, and 3pN with 100nM FHL2.
- #32. Figure S17 new figure. DISOPRED and PSIPRED predict N2B-us secondary structure.
- #33. Figure S18 new figure. AlphaFold2 predict similar structure of N2B-us structural domain in other organisms with human N2B-us structural domain.
- #34. Figure S19 new figure. In the predicted structure from AlphaFold2 the three cysteines are distal from each other.
- #35. Page 21 paragraph 2: Text S4 Rate of association for diffusion-limited binding added to discuss the binding of FHL2 to N2B-us structural domain in limited time interval.

B. Point-to-Point Responses to Reviewers' Comments

Reviewer #1 (Remarks to the Author):

In this paper, the authors use single-molecule manipulation by magnetic tweezers and AlphaFold2 predictions for structural and binding properties of a mechanically active region in cardiac titin, the N2B unique sequence (N2Bus). They confirm and extend their previous finding that regions in the N2Bus domain undergo unfolding-refolding transitions at low forces (between ~5 and ~15 pN) and identify an alpha/beta structural domain of ~115 AA which they characterize mechanically. They also confirm and extend the properties of N2Bus as a binding partner of Four-and-a-half-LIM domain protein 2 (FHL2), demonstrating that N2Bus has multiple binding sites for this protein. Binding affinity is increased by stretching N2Bus.

While the mechanical experiments using magnetic tweezers appear to be of high technical quality, several major issues greatly dampen my enthusiasm for this paper:

1. The authors' premises that the titin N2Bus has been considered an intrinsically disordered region is not really true in vivo: Several papers have shown that N2Bus is intrinsically structured by disulfide bonding. This has been reported by Grutzner et al, (Biophys J 97:825-34, 2009) and two recent papers using mass spectrometry to detect titin oxidation have demonstrated that N2Bus is oxidized in living heart muscle (Loescher et al, PNAS 117:24545-24556, 2020; Herrero-Galán et al, Redox Biol. 52:102306, 2022), likely reflecting the formation of disulfide bonds which cross-link this region and limit its extensibility. By contrast, the authors use 10 mM DTT to prevent any oxidation in their in vitro experiments, which raises the question of how physiologically relevant their findings are.

Response #1:

{

We would like to thank the reviewer for providing us with information regarding the oxidization of N2B-us and the resulting formation of disulfide bonds. Upon scanning the sequence of the predicted structural domain, we have identified three cysteines. Based on the structural prediction using AlphaFold2, these three cysteines are distal from each other, indicating that they are unlikely to form disulfide bonds once the domain is folded into its structure (see new Figure S19). Furthermore, these three cysteines are also distal from other cysteines in N2B-us, with the nearest one being spaced 104 residues upstream. Given these findings, it is unlikely that the formation of disulfide bonds between the transiently exposed three cysteines in the structural domain and those outside of the domain would be a kinetically favored process compared to the refolding of the domain under force.

To test whether the cysteines in the structural domain can form disulfide bonds easily, we performed additional experiments without DTT in the buffer. Our results showed that the structural domains continued to undergo repeated unfolding and refolding at similar forces, even in the absence of DTT. This suggests that the cysteines within the structural domain do not form disulfide bonds with each other or with cysteines

outside the domain when DTT is not present. These findings are consistent with our initial hypothesis and are shown in Figure S5.

In the revised manuscript, a few sentences are added to discuss the potential impact of the disulfide bonds on our experimental observations (page 8, paragraph 2, from line 15). The previous works mentioned by the reviewer have been cited (Refs. 34-36).
}

2. The unfolding-refolding transitions in N2Bus reported for the structural domain occur in a force range that is the same as that within which titin Ig domains unfold-refold (Eckels et al, *Annu Rev Physiol* 80:327-351, 2018). Because the elastic titin segment contains dozens of potentially unfolding-refolding Ig domains but only one N2Bus structural domain, it remains to be shown whether unfolding-refolding of the N2Bus structural domain occurs in muscle sarcomeres, in parallel with Ig-domain unfolding-refolding (which has been demonstrated in myofibrils by Rivas-Pardo et al (*Cell Rep* 14:1339-1347, 2016). At a minimum, the authors should develop a model which includes the mechanical properties of both the N2Bus and the 50+ I-band Ig-domains, to predict whether or not the unfolding of N2Bus may be functionally relevant in real sarcomeres with a full-length titin spring segment.

Response #2:

{

The reviewer asked us to conduct a simulation to understand how the other Ig domains in the I-band titin might affect the structural state of the N2B-us structural domain. Since the N2B-us structural domain can only detect local tension, the potential influence of the I-band Ig domains must be through their effects on the tension transmitted by the I-band titin. It is possible that force-dependent structural transitions of the I-band Ig domains may contribute to the regulation of tension levels, with more unfolded domains resulting in lower tension.

In our previous study, we simulated the I-band tension dynamics by considering the force-dependent unfolding and refolding of all the I-band immunoglobulin (Ig) domains under the assumption that they have similar mechanical properties to titin I27 (Pang, Le, & Yan, 2018). The simulation results suggest a tension level range of 1-5 pN during heart beating cycles, which is consistent with previous estimates by Linke et al. (Linke et al., 1999).

Therefore, our finding of the force-dependent unfolding of the N2B-us structural domain at forces above 4 pN (Figure 5) is physiological relevant.

A new paragraph on the physiological relevance of the force-dependent unfolding of the N2B-us structural domain is added to the Discussion section (page 18, paragraph 2 and page 19, paragraph 1-2). The papers mentioned by review have been cited (Ref. 50-51)

}

3. The proposed “mechanosensing” function of N2Bus caused by its stretch-dependent interaction with FHL2 is not convincingly shown. What the authors show is that more FHL2 binds to N2Bus (both the unfolded structural domain and other regions) when this segment is stretched. However, it is not clear how specific this effect is, because no control experiments are performed, e.g., with a protein that does not bind to titin but has a similar size as FHL2. Because N2Bus gets more sticky on stretching, due to the exposure of previously less accessible hydrophobic sites, the increased binding of FHL2 is not evidence of a relevant functional effect. Moreover, the authors do not really address a “mechanosensing” function of N2Bus, as they do not investigate the (potential) downstream effects of FHL2 binding-unbinding to N2Bus in a cardiac muscle cell.

Taken together, new insight into the physiological function of the FHL2-binding to stretched N2Bus titin is not provided.

Response #3

{

Following the constructive comments, we performed new experiments using two LIM domain containing proteins namely LIM domain only protein 3 (LMO3) and LIM/homeobox protein2 (LHX2), which are not reported to bind titin N2B-us. Adding 200 nM of these two proteins does not cause shift of the force-extension curve observed (Figure S12). Therefore, the observed FHL2 binding induced shift in the force-extension curve of N2B-us is not due to mechanical exposure of hydrophobic residues that lead to nonspecific binding of LIM domains in FHL2. We have mentioned this specificity in the main text (page 16 paragraph 4) and provided the data in figure S12.

Regarding the comment on the mechanosensing function of N2B-us via its binding to FHL2, we would like to draw the attention to a previous study showing that FHL2 binding to N2B-us region results in direct recruitment of creatine kinase, adenylate kinase and phosphofructokinase to titin (Lange et al., 2002). On the other hand, in its unbound state, FHL2 can target numerous other factors in cardiomyocytes. Since a few pN forces on N2B-us influence the balance between the bound and unbound equilibrium of FHL2, our results suggest mechanosensing of N2B-us via regulating the force-dependent binding of FHL-2, which might be related to the known important functions of FHL2 in heart. The mutations of FHL2 lead to cardiomyopathy (Friedrich et al., 2014). Mechanistically, FHL2 inhibits calcineurin pathway (Hojayev, Rothermel, Gillette, & Hill, 2012), Erk pathway (Purcell et al., 2004), suggesting that the force-dependent binding of N2B-usbinding factors may affect these functions.

}

Minor

1. The manuscript text would need substantial editing.

Response #4

```
{  
The manuscript has been significantly revised and proofread by our colleagues.  
}
```

2. (no page numbers are provided) Intro: The N2B element does not contain tandem-Igs, as stated wrongly. The proximal, middle and distal Ig-regions of titin do contain tandem-Ig segments, but not N2B.

Response #5

```
{  
The statement has been deleted.  
}
```

3. As stated above, the specificity of the FHL2 binding to N2Bus is not shown. Can the site(s) required for N2Bus-binding be identified within the FHL2 molecule? If these sites were mutated, would the reported effects be gone? The authors should use “sham” protein in control experiments (see major issues). They should also systematically use at least two different concentrations of FHL2, to look for a concentration-dependent effect.

Response #6

```
{  
As previously mentioned in response #3, we have performed control experiments to demonstrate the specific binding of FHL2 to N2B-us. We have also added experiments using varying concentrations of FHL2 protein, ranging from 20nM to 160nM (Figure S11). We have observed that as the concentration of FHL2 increases the rigidifying effect on N2B-us also increases. The data from these experiments are shown in the updated Figure S11 and discussed in the main text page 16 paragraph 3 line 6.  
}
```

4. The sequence of N2Bus is quite variable between species. Is the structural domain well preserved among humans and other vertebrates? Evolutionary preservation can be evidence of a preserved physiological function.

Response #7

```
{  
Following the question, we have run BLAST for human N2B-us structural domain sequence and found homologue sequences domains in other organisms including mouse, sheep, zebrafish, and rat, which shown high degree of similarity. AlphaFold2 prediction suggests that they all form similar structures containing three beta-strands and two helical domains (Figure S18). Page 21 paragraph 1 line 1.  
}
```

Reviewer #2 (Remarks to the Author):

This manuscript provides an exciting case study of how AlphaFold predictions can be used together with experimentation to gain mechanistic insight into how certain proteins function. Running AlphaFold on an intrinsically disordered region: N2B-us, the authors noticed that part of this region had a well predicted structural domain. Furthermore, additional regions were predicted to bind FHL2. These were experimentally confirmed using stretching experiments.

I don't have any major concerns and find this manuscript to be an exciting demonstration of how AlphaFold predictions can be used to generate testable hypotheses. But I do have a few minor concerns that can be easily addressed.

1) the methods (page 4) specify the uniprot IDs of the proteins (Q8WZ42 and Q14192) but the exact regions used as input to AlphaFold2 are not specified. On page 6, it's indicated that 572aa N2B-us sequence is used. Here it would be good to know what range was used (start and end positions).

Response #8

{

We are glad to hear that the reviewer considers this work exciting and thanks his efforts to review the manuscript. The sequences have been added to the materials & method part. Page 4 paragraph 1.

}

2) The authors claim that N2B-us is commonly considered intrinsically disordered (introduction, page 2) and cite a reference 14, but this reference (from 1999) seems rather old. I would suggest running N2B-us region through DISOPRED and PSIPRED (see: <http://bioinf.cs.ucl.ac.uk/psipred>) to see if this region is predicted as disordered and lacking of secondary structure. It would be interesting to see if AlphaFold was actually needed to make this prediction, or if it's something that could've been done with a more traditional method that looks at local conservation patterns.

Response #9

{

We would like to clarify that the term "the intrinsically disordered region" refers to a lack of stable tertiary structure but does not necessarily exclude the presence of secondary structural motifs. Based on the suggestion we received, we ran a prediction for the N2B-us region using DISOPRED and PSIPRED, which suggests the presence of secondary structural motifs such as strands and helices throughout N2B-us. However, DISOPRED and PSIPRED are not able to predict whether there is a tertiary structural domain in N2B-us. Therefore, we believe that AlphaFold2 is needed to predict the 115 amino acid structural domain. The results of this analysis can be found in the discussion section: Page 20, paragraph 2, line 2 and Figure S17. New references

DISOPRED and PSIPRED are added (Ref 45-46) correspondingly.}

3) Page 11 - "AlphaFold2 predicts FHL2 binding..." This is where PAE plots should be shown. PAE (predicted aligned error) predicts confidence of interaction between every pair of positions. Just because the final structures are close in 3D space does not necessarily mean that AF2 confidently "predicts" them as binding.

Response #10

```
{  
Following the suggestion, the PAE plot of FHL2 binding to N2B-us fragments and the  
corresponding description in the main text are added (Figure 3B, page 13, paragraph  
2, line 6).  
}
```

4) It would be interesting to see if the new version of AlphaFold2, specifically trained for predicting protein-protein interactions (AlphaFold-multimer), will also lead to the same observations between N2B-us and FHL2. This can be accessed here: <https://colab.research.google.com/github/sokrypton/ColabFold/blob/main/AlphaFold2.ipynb>

Response #11

```
{  
Thank you for the suggestion! We have tried the newer version of AlphaFold2 and  
found that it also shows similar binding. The structure of the complex and the PAE plot  
in new Figure 3A-B are based on the prediction from AlphaFold-multimer. The pdb files  
of the predicted structures are included in the archived file for resubmission.}
```

5) Page 20 - "has precisely predicted more than 98% protein in human genome". I don't think this statement is correct. According to AlphaFold abstract: "almost the entire human proteome (98.5% of human proteins). The resulting dataset covers 58% of residues with a confident prediction, of which a subset (36% of all residues) have very high confidence" - <https://www.nature.com/articles/s41586-021-03828-1>

Response #12

```
{  
The statement has been revised accordingly (page 24 paragraph 2 line 10).  
}
```

6) Figure S1 - Why only four of the 5 models are shown? It would be good to know which model is which, and their predicted confidence (plddt, ptm). Maybe color by plddt, as this would be a good indicator that the ordered region is of high confidence.

Response #13

```
{
```

All the five models colored by pI-DDT and corresponding PAE plots have been added to the new Figure S1.

}

7) Figure S2 - when describing "binding" the pae plots will be most informative, as these will provide how confident AlphaFold is about the interfaces observed.

Response #14

{

PAE plots have been added (new Figure S8).

}

8) I would suggest saving all the predicted outputs (all files) from AlphaFold and providing this an archive for readers. (If this wasn't already done?).

Response #15

{

The files have been provided as a single archived file during resubmission.

}

Reviewer #3 (Remarks to the Author):

The N2B region is generally thought to be an unstructured and elastic region of titin, and hence an important factor governing the passive elasticity of the entire titin protein. Therefore, these results demonstrating a structured and mechanically stable region within the N2B protein is very exciting. Despite this exciting finding, I believe that the data shown in the manuscript is predominantly 'representative traces', with very little analysis to confirm that these traces are indeed representative of a greater number of experiments. In fact, figures 1, 3 and 4 are showing representative traces with no quantification or analysis of the single molecule data. Figure 2 is the only figure with quantification, but panels B, C and D are not mentioned in the text of the manuscript. Furthermore, the authors do not uncover a mechanism explaining exactly what the FHL2 binding does to the structured region of the N2B-us. In the current form, I believe that the data presented in this manuscript is too preliminary for publication in Nature Communications.

Response #16

{

We are glad to hear that the reviewers consider the study exciting. Replies to each of his/her specific comments are provided below.

}

Specific comments:

1. In the manuscript the authors use the force spectroscopy hopping curves at 5.5pN

to infer that the protein has thermal stability. Recent work (J. Chem. Phys. 151, 185105 (2019); doi: 10.1063/1.5126071) has shown that it is not so simple to extrapolate thermodynamic information from these mechanical measurements. Therefore, if the authors want to state that this region of the N2B is also thermally stable, I would encourage them to do the experimental measurements. Or avoid merging the concepts of thermal and mechanical stability.

Response #17:

{

The stability assay we performed is based on the Boltzmann distribution of two states: the folded and the completely unfolded states, at a given force. This allows us to determine the free energy by analyzing the probability ratio of the two states, which is independent of the transition pathway as we do not consider the transition kinetics. To extrapolate the free energy difference to zero force, we must make assumptions about the force-extension curves of the two states. We assume that the folded state behaves like a freely rotating rigid body and the unfolded peptide is a randomly coiled polymer with a bending persistence length of 0.5-0.8 nm. This assumption may introduce some uncertainty, as we cannot exclude the possibility that at lower forces, the peptide may develop secondary structures before folding.

Following the suggestion from the reviewer, during the revision we performed additional experiment to measure the melting temperature of the folded structure using thermal shift assay. A sharp melting transition was observed at 62 degrees in Celsius indicating it is indeed a well folded, thermally stable structure (Figure 2B, Page 10 paragraph 2).

We have added a sentence to note the reader for possible uncertainty in the extrapolation to zero force free energy and referred to readers to the work mentioned by the reviewer (Page 10 paragraph 1 line 12). The paper reviewer mentioned has been cited (Ref 41)

}

2. In figure 1, there is no quantification of the three different constructs, merely a single trace. For example, is the unfolding force of the truncated form the same as the full length? Is the contour length of the truncated vs. full length protein the same? All constructs need to be fully characterized.

Response #18

{

In response to the comment, during the revision we quantified the unfolding force and the released contour length for the structural domain in the full-length N2B-us construct (revised Figure 2). The results show that the unfolding forces and released

contour lengths are similar for the isolated structural domain and the structural domain in the full-length construct. This confirms that the mechanical stability and size of the identified structural domain are not significantly affected by other regions of N2B-us. (Page 11, paragraph 1, line 2). More representative unfolding time traces from multiple tethers are provided in the Supplementary Materials (Figure S2-S4).

}

3. In figure 2, only panel A is mentioned in the text, the other results are not mentioned. Furthermore, there are details missing in the caption. For example, what is the fitting curve in panel c? Furthermore, figure 2 is the quantification of a trace shown in figure 1 and I believe that these figures can be combined into 1.

Response #19

{

We have added detailed description in the main text for this figure (Page 11, paragraph 1). The curve is the theoretical prediction of the force-extension curve of the structural domain. The trace in figure is for comparison between different groups, while the traces in figure 2 are representative curves of the loading force assays. We prefer to keep both for clarity.

}

4. In the text, the authors state that there are 23 binding sites for FHL2 LIM domains, but in the scheme of binding sites in figure3, I count 24 binding sites represented.

Response#20

{

Many thanks for spotting the mistake. The number of total binding sties should be 24, which has been corrected in the main text. (Page 3 paragraph 3 line 6)

}

5. It is not clear to me exactly what is shown in the force – bead height curves in figure 3. Are these a single protein before and after the addition of FHL2? Or this is the combined data of many measurements? From the text I have understood that each curve is a single protein. If so, I believe the authors should not show a single protein and should show the analysis from many different experiments to show consistent changes in behavior upon the addition of FHL2.

Response#21

{

The representative force-bead heigh curves in each panel of figure 3 D-F are from a single tether before and after introducing FHL2, so the readers can clearly see the change caused by FHL2. Following the suggestion, to demonstrate the reproducibility of such FHL2-dependent changes in the force-bead height curves, we have summarized data from multiple independent tethers in the new figure panels Figure

3H-J. The data show that the bead height become significantly higher after adding FHL2 in 1.5 and 5pN, suggesting binding induces stiffening of the bound sites on N2B-us. While in 18pN full-length N2B-us and N2B-us ΔS constructs have lower bead height after adding FHL2, suggesting that the binding of FHL2 causes a deformation (looping or bending) of the binding site that leads to a reduction of effective contour length of the N2B-us.

}

6. Are the differences statistically significant with multiple molecules / multiple experiments? There should be controls with the addition of something that does not bind to N2B – ideally a mutated form of FHL2 that has no binding affinity to the N2B domain.

Response#22

{

Thank you for the question and the suggestion of additional control experiments. We have performed multiple experiments to quantify the difference in bead height before and after adding FHL2, and the results indicate a significant difference (Figure 3 H-J). To set the control group for the experiment, we used two LIM domain-containing proteins, LMO3 and LHX2, which have not been reported to bind Titin. The addition of LMO3 and LHX2 at a concentration of 200 nM did not cause any noticeable shift in the force-bead height curves (Figure S12, page 16, paragraph 4). Taken together, these results suggest that the force-dependent binding of N2B-us to FHL2 is specific.}

6. In figure3, panel E, the N2B-us- ΔS shows hysteresis without FHL2. But in the supplementary figures, it is the N2B-us-S construct that shows hysteresis. Why are there these differences?

Response# 23

{

Many thanks for spotting the mistake due to a labelling error, which has been corrected in the revised manuscript (new figure 3 EF).

}

7. It is not clear exactly what effect FHL2 has on the mechanical stability and folding behaviour of the N2B region. Are there different mechanisms depending on where FHL2 binds?

Response# 24

{

N2B-us can be divided into two regions, the ~115a.a. structural domain, and the spanning disorder region. As predicted by AF2 and confirmed by our study, both regions provide FHL2 binding sites.

The binding of FHL2 to the disorder region stiffens the binding sites, indicated by increased extension at a few pN forces (figure 3 F J). The binding of remote binding sites may cause looping of certain length of N2B-us disorder regions, suggested by shorter extension after binding at forces greater than 8 pN.

The binding of FHL2 to the structural domain requires mechanical unfolding of the domain to expose the cryptic binding sites. Once these binding sites are exposed and bound by FHL2, the binding of FHL2 will inhibit the refolding of N2B-us structural domain (Figure 4 AB, figure 5 G).

Together, FHL2 binds two different regions of N2B-us and has different effect: Stiffening the peptide in the disordered region over a few pN forces; Suppressing refolding of the structural domain after binding to the unfolded domain.

}

8. In figure4 the authors say that they have an upper limit of force of 6 pN to not induce rapid unfolding of the protein upon addition of FHL2. I would suggest that if this is the case, that the authors do not perform force-ramp experiments, and instead perform these experiments in force clamp conditions. If the authors held the protein at the hopping force and see an equal distribution between the unfolded and folded state, then upon addition of FHL2, if there is a change in the protein conformation this will alter the probability of the protein residing in one of the states. These experiments would be much more informative than these incomplete force-ramp protocols.

Response# 25

{

Following the suggestion to perform force-clamping experiments to demonstrate binding. We have performed such experiments and demonstrated binding at 4.5 pN (new figure 5, page 19, paragraph 2).

}

Friedrich, F. W., Reischmann, S., Schwalm, A., Unger, A., Ramanujam, D., Munch, J., . . . Carrier, L. (2014). FHL2 expression and variants in hypertrophic cardiomyopathy. *Basic Res Cardiol*, 109(6), 451. doi:10.1007/s00395-014-0451-8

Hojayev, B., Rothermel, B. A., Gillette, T. G., & Hill, J. A. (2012). FHL2 binds calcineurin and represses pathological cardiac growth. *Molecular and cellular biology*, 32(19), 4025-4034. doi:10.1128/MCB.05948-11

Lange, S., Auerbach, D., McLoughlin, P., Perriard, E., Schäfer, B. W., Perriard, J.-C., & Ehler, E. (2002). Subcellular targeting of metabolic enzymes to titin in heart muscle may be mediated by DRAL/FHL-2. *Journal of cell science*, 115(Pt 24), 4925-4936. Retrieved from <https://pubmed.ncbi.nlm.nih.gov/12432079>

Linke, W. A., Rudy, D. E., Centner, T., Gautel, M., Witt, C., Labeit, S., & Gregorio, C. C. (1999). I-band titin in cardiac muscle is a three-element molecular spring and is critical for maintaining thin filament structure. *The Journal of cell biology*, 146(3), 631-644.

- Pang, S. M., Le, S., & Yan, J. (2018). Mechanical responses of the mechanosensitive unstructured domains in cardiac titin. *Biology of the cell*, *110*(3), 65-76. doi:10.1111/boc.201700061
- Purcell, N. H., Darwis, D., Bueno, O. F., Müller, J. M., Schüle, R., & Molkenin, J. D. (2004). Extracellular signal-regulated kinase 2 interacts with and is negatively regulated by the LIM-only protein FHL2 in cardiomyocytes. *Molecular and cellular biology*, *24*(3), 1081-1095. Retrieved from <https://pubmed.ncbi.nlm.nih.gov/14729955>

Reviewers' Comments:

Reviewer #2:

Remarks to the Author:

The authors have addressed my concerns. I would recommend for publication! It is exciting to see the PAE plots.

Minor comments:

- In figures S11, S12, the y-axis is not shown (I think there might be an issue in how the was embedded in the PDF).

Reviewer #3:

Remarks to the Author:

In the amended manuscript entitled " Mechanical regulation of Titin N2B-us conformation and its binding to FHL2" by Sun et al, the authors have performed additional experiments and analysis, including both melting temperature assays and single molecule magnetic tweezer experiments to strengthen the manuscript.

While I thank the authors for the effort and time to improve the manuscript, I still do not feel that there is sufficient mechanistic evidence to support the claims made in this manuscript. For example, the authors state that their "experiments revealed that FHL2 has a high affinity for N2B-us and binds to it in a highly regulated manner, influenced by mechanical stretching" but it is not clear to me how this binding is highly regulated or what are the regulation mechanisms. Furthermore, I feel that the evidence shown by the authors that the FHL2 binding is force activated (mainly demonstrated in figure4) is lacking in depth quantitative analysis.

While I still believe that the authors identification of the stable domain in the N2B region is interesting, the emphasis of the paper is placed on the interaction between the N2B and FHL2 domain. I do not feel that the mechanisms regulating this binding interaction have been fully described.

Other comments:

Figure captions need expanding, at present there are parts of the figures that are unclear and are not described adequately in the captions. For example in figure 3c, there is a blue and orange box with a-helix and b-sheet written inside – it is not clear what this box is meant to represent, whether it is the FHL2 or the N2B domain. Or in figure 3B, the colour map is from 0-30, but without a label or units, is this is the predicted error in angstrom?

The authors have performed single molecule magnetic tweezers experiments at 37degC, but a description of how the temperature is controlled in their set-up is not described in the materials and methods.

In the caption of figure 3, it is written "(B). predicted alignment error plot if the predicted structure.". This sentence needs re-writing.

Supplementary figure 12 has the figure letters and axis cut off. Furthermore, the description of panel b "No binding induced curve shifting observed neither" needs re-writing.

List of Main Changes and Point-to-Point Responses

List of Main Changes:

Main Changes in Main Text:

1. We changed the title to "Discovery of a Structural Domain in the Titin N2B-us Region that Binds to FHL2 in a Force-Activation Dependent Manner."
2. We added a new results section: "Mechanical Regulation of Full-Length N2B-us Binding to FHL2." In this section, two new experiments were used to demonstrate the force regulation of N2B-us binding to FHL2.
3. We changed the presentation order, shifting the results section to before the methods section, to make it consistent with Nature Communications style.
4. We have rewritten the "FHL2 LIM Domain Binding Sites Spread Throughout the N2B-us" section to make it more logical. We also moved some remarks to the discussion.
5. We have rewritten the "Mechanical Activation of N2B-us Structural Domain's Binding to FHL2" to improve the logic.
6. We removed the results section "N2B-us Structural Domain Unfolds and FHL2 Binding Can Happen Under Physiological Conditions" and added the content to other results sections.
7. We deleted the discussion about combining AF2 and single molecule manipulation, given that it has been widely used now. Several discussion paragraphs were rewritten to make them clear and logical.
8. All changes were marked as red in the manuscript file.

Main Changes in Figures:

1. Figure 1: Changed the illustration of panel A. Some captions were rewritten.
2. Figure 2: Captions were rewritten.
3. Figure 3: T-test results were added to figures G-I. Captions were rewritten.
4. Figure 4: New data obtained from a force-jump binding assay, which shows that FHL2 binding N2B-us is facilitated by forces of a few pN.
5. Figure 5: New data obtained from a force-dependent fluorescence imaging assay, which consistently shows that FHL2 binding N2B-us is facilitated by forces of a few pN.

6. Figure 6: Revised from Figure 4 in previous submissions. Panels C & D are newly added.

Main Changes in Supplementary Information:

1. Resized the text and figures in the supplementary information to enhance their aesthetic appeal and re-sequenced them for sequential referencing in the main text.
2. Updated several supplementary materials, including Figures S8, S14, S15, S19, and Text S3.
3. Removed Texts S2 and S4 from the supplementary materials.

Other minor revisions are also marked in red.

Point-to-point responses to reviewers' comments

Reviewer #2 (Remarks to the Author):

The authors have addressed my concerns. I would recommend for publication! It is exciting to see the PAE plots.

Minor comments:

- In figures S11, S12, the y-axis is not shown (I think there might be an issue in how the was embedded in the PDF).

Response 1:

{

We thank for the reviewer for the recommendation for publication. The figure S11, S12 have now been reordered as figure S12 and figure S13 in the revised manuscript. The y-axis is now properly shown.

}

Reviewer #3 (Remarks to the Author):

In the amended manuscript entitled “Mechanical regulation of Titin N2B-us conformation and its binding to FHL2” by Sun et al, the authors have performed additional experiments and analysis, including both melting temperature assays and single molecule magnetic tweezer experiments to strengthen the manuscript.

While I thank the authors for the effort and time to improve the manuscript, I still do not feel that there is sufficient mechanistic evidence to support the claims made in this manuscript. For example, the authors state that their “experiments revealed that FHL2 has a high affinity for N2B-us and binds to it in a highly regulated manner, influenced by mechanical stretching” but it is not clear to me how this binding is highly regulated or what are the regulation mechanisms. Furthermore, I feel that the evidence shown by

the authors that the FHL2 binding is force activated (mainly demonstrated in figure4) is lacking in depth quantitative analysis.

While I still believe that the authors identification of the stable domain in the N2B region is interesting, the emphasis of the paper is placed on the interaction between the N2B and FHL2 domain. I do not feel that the mechanisms regulating this binding interaction have been fully described.

Response 2:

{

We thank reviewer for the careful review and the valuable comment to help improve the quality of our manuscript.

To address the concerns from the reviewer, we have conducted two new types of experiments to show that the binding of FHL2 is highly regulated by a few pN forces. We have also discussed the regulation mechanisms based on force-dependent exposure of FHL2's LIM domain binding sites in the structural domain identified in this work.

New experiments

Force-jump assay: The first type of experiment was a force-jump magnetic tweezer experiment on N2B-us. In this experiment, we applied force to N2B-us using two different procedures. The first procedure involved holding an N2B-us at 1 pN for 10 seconds, then increasing the force to 8 pN for 20 seconds, and finally returning to 1 pN for 10 seconds. We determined the difference in bead heights at 1 pN by comparing measurements during the initial 10 seconds with those in the last 10 seconds. We observed an approximately 10 nm increase in height in the presence of 200 nM FHL2, while a nearly zero nm height difference was observed in the absence of FHL2. This indicates that the approximately 10 nm height increase in the presence of 200 nM FHL2 was due to the binding of FHL2 to N2B-us.

In another procedure, we examined whether the transient exposure of N2B-us to 8 pN was necessary for FHL2 binding. A constant 1 pN force was applied for 40 s, and the difference in bead heights recorded during the first 10 s and the last 10 s was determined. No significant bead height difference was observed, regardless of whether FHL2 was present. Therefore, we can conclude that the 8 pN exposure in the first procedure was responsible for the observed FHL2 binding.

Based on other data reported in this manuscript (Figure 2A), the structural domain in N2B-us remains folded at 1 pN and has a large chance of unfolding at 8 pN. The mechanical unfolding of this domain exposes the cryptic FHL2's LIM domain binding sites, allowing binding by FHL2. This binding could subsequently suppress the refolding of the domain when the force is reduced back to 1 pN. This mechanism is

likely a major factor accounting for the observed height difference. Additionally, FHL2's binding to regions outside the structural domain may also contribute to a certain level of bead height change. This experiment was added to the manuscript in figure 4.

Single-molecule fluorescence imaging experiment: The second type experiment involved single-molecule fluorescence in combination with magnetic tweezers. We applied low (~ 1 pN) or high (~ 10 pN) forces to N2B-us while using a TIRF microscope to observe quantum dot-labeled FHL2. For each N2B-us tether, it was first exposed to 1 pN for approximately 2 minutes, and then the force was increased to 10 pN for another 2 minutes. Only when FHL2 binds to N2B-us can it remain in the evanescent field and emit a stable signal. The region of interest was chosen to be right below the DNA handle, based on the signal from the fluorescent DNA dye (Sytox-orange).

In such experiments, 19 independent tethers were involved. When the N2B-us tethers were held at ~ 1 pN, the Quantum dot fluorescent intensity remained constant low. In sharp contrast, when ~ 10 pN force was applied, signal intensity was significantly higher, implies that the high force facilitates binding of FHL2. We also used quantum dots without FHL2 as control, which demonstrated that the signal remains low under both high and low force conditions. This part of the experiment is presented in figure 5 of the manuscript.

Together, these two types of experiments consistently suggest an enhanced binding of FHL2 to N2B-us with a few pN forces.

Data analysis

Additionally, we agree with reviewer#3's suggestion that our data requires more quantitative analysis. We have re-analyzed both our existing and new datasets, presenting them with enhanced quantitative and statistical rigor to determine their statistical significance (Figure 3,4,5,6).

We also concur with reviewer #3's interest regarding the novel domain identified in our study. Reflecting this, we propose to retitile the manuscript as "Discovery of a Novel Structural Domain in the Titin N2B-us Region that Binds to FHL2 in a Force-Activation Dependent Manner," emphasizing the novelty of discovering the structural domain.}

Other comments:

Figure captions need expanding, at present there are parts of the figures that are unclear and are not described adequately in the captions. For example in figure 3c, there is a blue and orange box with a-helix and b-sheet written inside – it is not clear what this box is meant to represent, whether it is the FHL2 or the N2B domain. Or in figure 3B, the colour map is from 0-30, but without a label or units, is this is the predicted error in angstrom?

Response 3:

{

We have extended figure caption for figure3C to clarify that the helical and sheet region belong to N2B-us structural domain. In addition, we also changed the illustration to make it easier to understand. We have also substantially edited all the figure captions, including those in the supplementary figures, to make them more precise and clearer. The edited figure captions were marked as red color in the manuscript.

The color map in Figure 3B indeed represents the Predicted Aligned Error (PAE) in angstroms. In AlphaFold2, PAE provides a distance error for every pair of residues, covering all possible combinations within the structure. That means, it offers an estimate of the position error at residue x when the predicted and true structures are aligned on residue y. We have added this explanation to the main text (page 6 paragraph 2) and also modified the figure and caption to give better explanation (Figure 3B).

}

The authors have performed single molecule magnetic tweezers experiments at 37degC, but a description of how the temperature is controlled in their set-up is not described in the materials and methods.

Response 4:

{

We employed a temperature control system with a heating film attached to the channel. We have added details to method section and cited the paper with more detailed description (page 15, “Single-molecule manipulation section, second last paragraph”)

}

In the caption of figure 3, it is written “(B). predicted alignment error plot if the predicted structure.”. This sentence needs re-writing.

Response 5:

{

We have rewritten the sentence as “(B). Predicted aligned error (PAE) plot for the predicted structure. The area corresponding to binding interface shows low error (blue), suggesting that the prediction for the binding interface is confident.”(Figure 3B).

}

Supplementary figure 12 has the figure letters and axis cut off. Furthermore, the

description of panel b “No binding induced curve shifting observer neither” needs re-writing.

Response 6:

{

The figure S12 is renumbered to figure S13 in our new version. The picture has been recovered and the caption has been corrected (Figure S13).

}

Reviewers' Comments:

Reviewer #3:

Remarks to the Author:

I appreciate that the authors have done a considerable amount of work to improve both the science and flow of the manuscript.

My concerns have been satisfied.

Reviewer #3 (Remarks to the Author):

I appreciate that the authors have done a considerable amount of work to improve both the science and flow of the manuscript.

My concerns have been satisfied.

We thank the reviewer for the comments and support for publication.